# Multi-molecular hyperspectral PRM-SRS microscopy

Wenxu Zhang[1,9], Yajuan Li[1,9], Anthony A. Fung[1,9], Zhi Li[1], Hongje Jang [1], Honghao Zha[1], Xiaoping Chen[2], Fangyuan Gao[3], Jane Y. Wu[2], Huaxin Sheng[4], Junjie Yao [5], Dorota Skowronska-Krawczyk [3], Sanjay Jain [6,7,8] & Lingyan Shi[1] ✉

Lipids play crucial roles in many biological processes. Mapping spatial distributions and examining the metabolic dynamics of different lipid subtypes in cells and tissues are critical to better understanding their roles in aging and diseases. Commonly used imaging methods (such as mass spectrometry-based, fluorescence labeling, conventional optical imaging) can disrupt the native environment of cells/tissues, have limited spatial or spectral resolution, or cannot distinguish different lipid subtypes. Here we present a hyperspectral imaging platform that integrates a Penalized Reference Matching algorithm with Stimulated Raman Scattering (PRM-SRS) microscopy. Using this platform, we visualize and identify high density lipoprotein particles in human kidney, a high cholesterol to phosphatidylethanolamine ratio inside granule cells of mouse hippocampus, and subcellular distributions of sphingosine and cardiolipin in human brain. Our PRM-SRS displays unique advantages of enhanced chemical specificity, subcellular resolution, and fast data processing in distinguishing lipid subtypes in different organs and species.

Current lipidomic technologies, such as shotgun lipidomics, can quickly identify hundreds of lipids from small samples. Albeit highly sensitive, such methods rely on mass spectrometry (MS), nuclear magnetic resonance (NMR), or other techniques that are destructive to cells and tissues[1–5]. Conventional matrix-assisted laser desorption/ionization (MALDI)-MS imaging enables label-free lipid imaging but it has a lateral resolution on the order of cell diameters (~10 μm) and destroys the sample during the imaging process. In addition, 3D MALDI imaging relies on serial sections of the sample, and the lipid species that are resolvable are limited to those with the highest ion yields. Other optical techniques have been developed to non-destructively visualize spatial distributions of lipid subtypes as well as metabolic flux[6] at the subcellular resolution, but they rely on markers such as fluorescently labeled antibodies and transfected biosensors, which may alter the native distribution of lipids in cells or tissues. It is difficult to use labeled optical imaging to differentiate diverse molecular species simultaneously, since the diversity of lipid species far exceeds the specificity and availability of optical tags and dyes. Therefore, label-free optical imaging is instrumental. Stimulated Raman scattering (SRS) microscopy has demonstrated the advantages of non-destructive 3D imaging with subcellular resolution in a label-free manner[7,8]. Recent work has even demonstrated quantitative mass concentration measurements of lipids, proteins, and water[9]. For label-free SRS imaging microscopy, multiple subcellular organelles and their chemical compositions can be visualized and mapped out through hyperspectral imaging (HSI) or training of a deep learning

[1]Shu Chien-Gene Lay Dept. of Bioengineering, University of California San Diego, La Jolla, CA, USA. [2]Dept. of Neurology, Northwestern University School of Medicine, Chicago, IL, USA. [3]Center for Translational Vision Research, School of Medicine, University of California Irvine, Irvine, CA, USA. [4]Dept. of Anesthesiology, Duke University School of Medicine, Durham, NC, USA. [5]Dept. of Biomedical Engineering, Duke University, Durham, NC, USA. [6]Dept. of Medicine, Washington University in St. Louis, St. Louis, MO, USA. [7]Dept. of Pathology & Immunology, Washington University in St. Louis, St. Louis, MO, USA. [8]Dept. of Pediatrics, Washington University in St. Louis, St. Louis, MO, USA. [9]These authors contributed equally: Wenxu Zhang, Yajuan Li, Anthony A. Fung. ✉e-mail: Lshi365@gmail.com

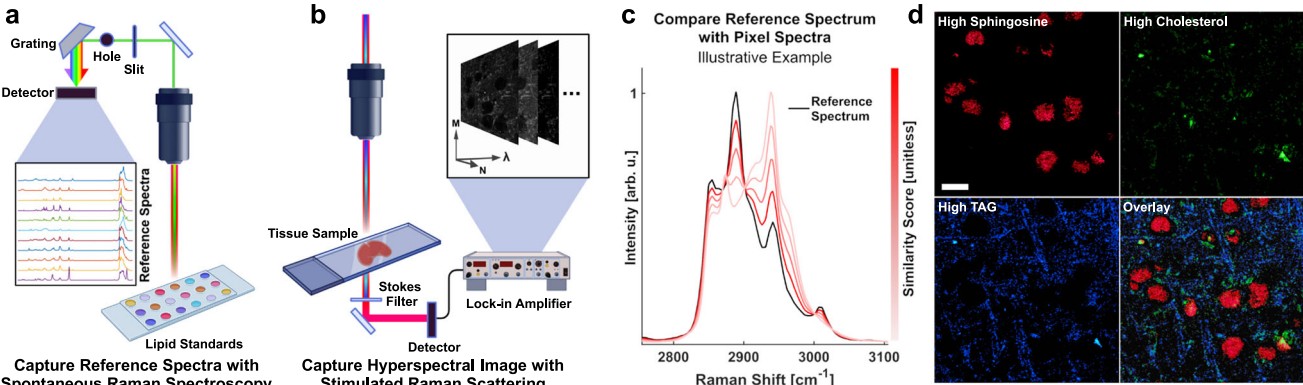

**Fig. 1 | General Reference Matching Method. a** A lipid subtype standard is analyzed by spontaneous Raman spectroscopy and preprocessed to generate a reference spectrum. **b** A sample is imaged using SRS to generate a HSI. **c** Each pixel of the HSI is a vector of intensity values that represent the Raman spectrum at that pixel. These spectra are compared using spectral angle mapping and illustrate how dissimilar spectra have a lower cosine similarity. **d** An example of a mouse brain sample with thresholded similarity scores with respect to sphingosine, cholesterol, and TAG. Pixel intensities are scaled to their similarity scores. SRS Stimulated Raman Scattering, HSI Hyperspectral Image, TAG triacylglyceride. (Panels (**a**) and (**b**) were created with BioRender.com).

model[10]. Lock-in free multiplex SRS imaging can rapidly extract hundreds of morphological or metabolic features in situ to understand lipid metabolism in cancer cells[11]. Despite these advancements, there has been no report on distinguishing multiple lipid subtypes in cells and tissue samples by using nondestructive label-free optical imaging methods.

In addition to imaging technologies, post-processing methods/algorithms also contribute to producing high-resolution and high-quality images. Recent work on Raman HSI analysis using multivariate curve resolution alternating least squares (MCR-ALS) algorithm has demonstrated effective unmixing of chemical species without disturbing the native distribution of biomolecules[12]. However, a higher spectral resolution may entail prohibitively long imaging time. In addition, unmixing lipid species using unsupervised methods can be computationally expensive and lack the ability to directly identify a chemical species without manual association *posteriori*. For example, the MCR-ALS approach converts a complex spectrum to a linear combination of component spectra, but it can take 30 min to process a 512 × 512-pixel hyperspectral image and presupposes the number of chemical species in a sample. The result displays a pixel's identity by its relative proportional composition of reference species. However, this is not always feasible in a complex biological sample. Singular Value Decomposition (SVD) can estimate the number of components, but analytical results may be sensitive to slight deviations from the exact number of components. Clustering and segmentation of image pixels may be informed by MCR-ALS, however, the precise molecular identities of the highlighted pixels may still be unknown, as there is no guarantee that the unmixed components correspond to a specific molecular type.

Spectral reference matching approaches, also known as spectral angle mapping, have been widely applied to Raman spectral analyses by quantifying the spectral similarity between an image pixel spectrum and a known reference spectrum[13]. Figure 1 shows the general process of reference matching approach applied to hyperspectral imaging. First, spectra of the target analytes (the reference standards of interest) are acquired using spontaneous Raman spectroscopy (Fig. 1a) and preprocessed for background removal and normalization (see Methods for details). Hyperspectral Raman microscopy imaging is next performed on the sample of interest, in which each pixel contains specific spectral information (Fig. 1b). Then each pixel's spectrum is preprocessed in the same way as the reference spectrum and is analyzed with respect to the reference spectrum by calculating the cosine similarity score (Fig. 1c). By repeating this step pixel-by-pixel, resulting images representing the dominant lipid subtypes are generated (Fig. 1d). However, the general spectral reference matching approach has low specificity and the high incidence of false positives makes it difficult to implement in vibrational spectroscopy. This is because the peak position and intensity differences of spectra generated by various equipment can produce uncertainty that overshadows the subtle differences between lipid subtypes.

To enhance the specificity for distinguishing lipid species accurately, in this work, we develop a Penalized Reference Matching (PRM) algorithm and apply it to SRS (PRM-SRS) microscopy. We focus on using dominant components to illustrate the application of PRM-SRS in analyzing different lipid subtypes in a variety of organs and species. We accumulate a library of 38 biomolecules for potential detection. This method is efficient and can process a 512 pixels × 512 pixels × 76 hyperspectral image stack within one minute. In future follow-up studies, we will further enhance the detection sensitivity to improve the signal-to-noise ratio for examining molecules of low abundance. This method will provide quantitative and qualitative insights into different roles of lipid species in multiple biological processes and can augment other unmixing techniques as well.

## Results

### Developing a penalized reference matching (PRM) method

SRS HSI pixel spectra and reference spectra were linearly interpolated such that the spectral interval is 1 cm$^{-1}$ wavenumber. This ensures that the inner product, which requires vector dimensions to be the same, is possible. After all spectra were adjusted to the same interpolated resolution, they were simplex normalized using Eq. 1,

$$I_1 = \frac{I - I_{min}}{I_{max} - I_{min}} \tag{1}$$

where $I_{min}$ is the minimum value and $I_{max}$ is the maximum value in the pixel spectrum. This normalization is done prior to reference matching so that the process generates results that are solely based on spectral shape without being affected by any intensity fluctuation. The normalized pixel spectra were then divided by their Euclidean norm as shown in Eq. 2

$$I_2 = \frac{I_1}{||I_1||_2} \tag{2}$$

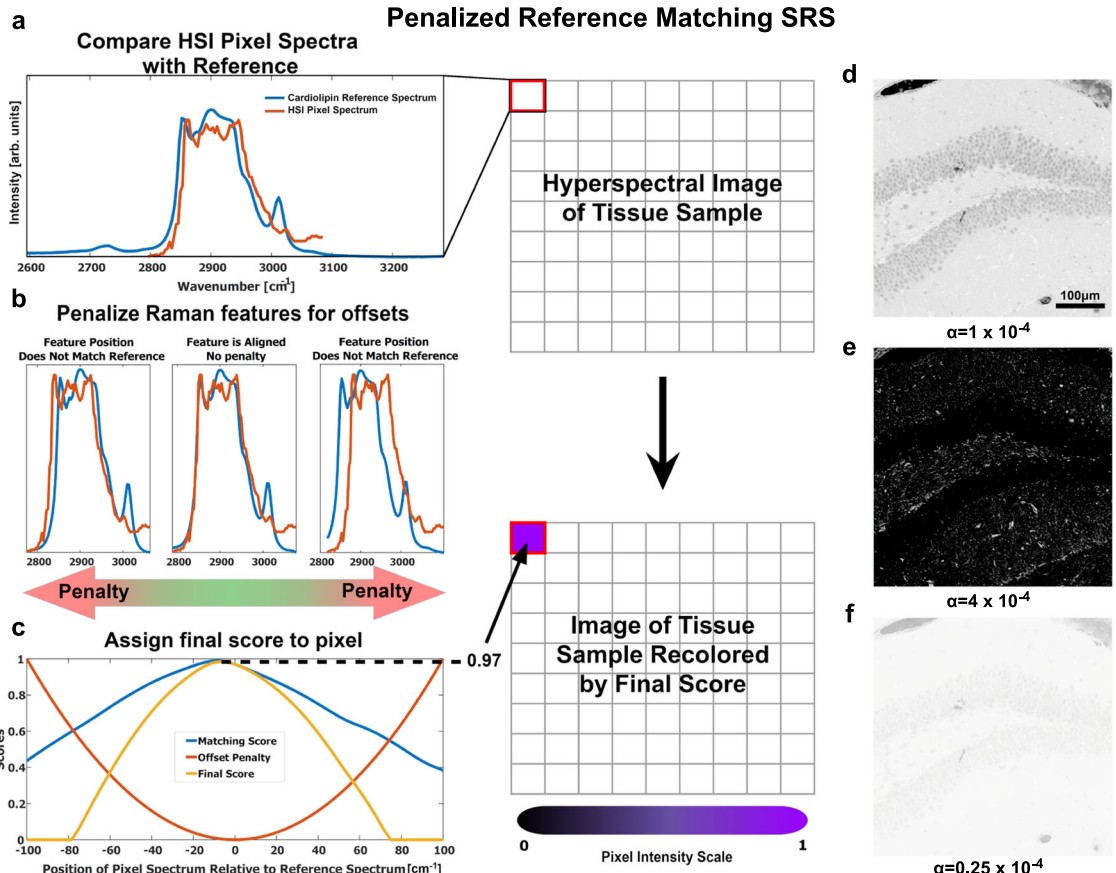

**Fig. 2 | Penalized Reference Matching Method. a** A lipid reference spectrum (blue) was collected by spontaneous Raman spectroscopy, and an SRS HSI pixel spectrum (orange) was compared with the reference spectrum. **b** Demonstration of shifts in pixel's spectrum: 7.13 cm$^{-1}$ left shift and 21.39 cm$^{-1}$ right shift. These positional offsets decrease the final similarity score because the penalty term scales exponentially to the positional offset. **c** Matching scores are retrieved by dot product between the reference signal and SRS pixel's signal. Then the penalty, estimated with a quadratic function, is subtracted from each matching score. When the shift wavenumber is high, a high penalty will be given. The highest value in this score curve will be used as the similarity score between this pixel spectrum and that of the pure reference standard. **d** Image illustrating the distribution of cardiolipin in a murine dentate gyrus sample with a penalty coefficient of $\alpha = 1 \times 10^{-4}$. **e** Image with the penalty coefficient $\alpha = 4 \times 10^{-4}$. The full range of the cardiolipin distribution was not clearly shown due to over-penalization. **f** An over-saturated image with the penalty coefficient $\alpha = 0.25 \times 10^{-4}$. Almost all pixels have a high similarity score due to under-penalization. Scale bar, 100 μm. HSI Hyperspectral Image.

where $I_2$ is the interpolated signal of the pixel spectrum. Reference spectra from spontaneous Raman acquisitions follow the same preprocessing steps as the HSI pixel spectra, as shown in Eqs. 3 and 4 below.

$$I_4 = \frac{I_3 - I_{3,\min}}{I_{3,\max} - I_{3,\min}} \tag{3}$$

$$I_5 = \frac{I_4}{\|I_4\|_2} \tag{4}$$

where $I_3$ denotes spontaneous Raman spectra, and $I_5$ is the interpolated signal of the reference spectrum. Due to the nature of Raman spectral intensity, similarity scores between each pixel spectrum and the reference spectrum were calculated using the dot product of $I_2$ and $I_5$.

These spectroscopic methods have been deployed for several decades, but due to high false positive rates, direct label-free characterization of multiple lipid subtypes in cells and tissues has not been achieved by using optical imaging. To address this, we added a penalty term to the canonical cosine similarity algorithm, which decreases the false positive rates by proportionally reducing the similarity score with the positional discrepancy to the best spectral match (Fig. 2a–c). This process is summarized as

$$\text{score} = (\mathbf{u_i} \cdot \mathbf{v} - \alpha \triangle x_i^2) \tag{5}$$

where $\mathbf{u}$ represents the interpolated signal of a pixel's shifted spectrum at various positions; $\mathbf{v}$ represents the interpolated signal of the reference spectrum; $\alpha$ is the penalty coefficient, with a unit of [cm$^2$]; $\Delta x_i$ is the deviation in position of the spectrum in $\mathbf{u_i}$ from the initial observed position; and $N$ is the number of interpolated signals, which depends on the spectral resolution of the HSI. The penalty term $\alpha\Delta x^2$ inherently addresses the slight positional deviations due to the diverse chemical environment, as well as the variations in instrumentation (such as thermoelectric noise, lensing, and other interference). Without this term, even if the spectral shape of a pixel matches the reference spectrum (Fig. 2a), the final similarity score may still be low when the positions of the peaks differ greatly (Fig. 2b, c). With the penalty term, all pixel spectra are evaluated as they occur at multiple Raman shifts, and the highest similarity score is returned in a pixel-by-pixel manner.

By leveraging positional information in addition to peak shape, the breadth of similarity score is increased or decreased, akin to a change in contrast (Fig. 2d–f). This ensures that pixels with similar shapes and positions are scored accordingly.

Most images collected in this study were taken from the Raman CH stretching region (2700–3150 cm$^{-1}$) with 75 total Raman shifts

(a spectral distance of 6 cm$^{-1}$ between images). The position deviation $\Delta x$ was the shift of peaks in the spectrum. We assessed several values for the penalty coefficient and chose $\alpha = 1 \times 10^{-4}$. At a higher value ($4 \times 10^{-4}$) of $\alpha$, the image contrast was too high to show the full-range signals, whereas a lower $\alpha$ ($0.25 \times 10^{-4}$) caused over-saturation in images (Fig. 2d–f). This is because if the penalty is too low, the pixel and reference spectra are free to shift themselves relative to each other until the highest similarity score is returned, no matter how far that shift may be from the original position. In addition, to show the capability of analysis based on CH stretching region signal only, we fitted the spectra of 38 lipid subtypes using four Gaussian peaks corresponding to functional groups defining lipid structures as shown in Supplementary Fig. 1a and Table 1. The fitting parameters were listed in Supplementary Table 2, and t-SNE plot of the parameters shows the spectral differences between different lipid subtypes.

Since Raman spectra contain molecular bond information that correlates with concentration, similarity scores may be used to estimate the relative levels of different molecules, such as different lipid subtypes. When the Raman spectra of a sample exhibit a high degree of similarity to that of a reference standard, it will suggest a higher concentration of that reference molecule in the sample. Different biomolecules may have the same types of chemical bonds, and the cumulative mixture of various molecules may result in a spectrum that displays the same spectral shape as an unrelated molecule. From a macromolecular perspective in biological samples, however, we find that factors such as the diversity of the analyte composition do not necessarily void the correlation between relative ratios and similarity scores (Supplementary Figs. 2 and 3, also see data presented later in Fig. 7).

We further compared PRM with pseudo-inverse matrix (PINV) multiplication and found that both PRM-based similarity scores and PINV-derived coefficients correlated with relative ratios of lipids. However, PINV-derived coefficients can have negative values and have unbounded ranges. Furthermore, calculation using PINV took much longer time than PRM (Supplementary Figs. 3, 4a, b). Even though PINV coefficients intuitively classify the similarity of random spectra as having no similarity (a value of 0), a perfect match may not have a coefficient of 1 (Supplementary Fig. 4b). Since the Raman spectra of biological samples are not entirely random, and share some general similarities, it is more important that perfect matches be bound to a value of 1, even if the similarity scores of random spectra are not centered at 0. Furthermore, HSIs are often acquired at various pixel densities and resolutions, which yield datasets of different sizes. Whether a 512 × 512 px or a 256 × 256 px HSI is acquired for the same sample, the pixels corresponding to the same structures should have the same spectrum, and therefore the same similarity score. However, since PINV coefficients for the spectra in a HSI are not mutually exclusive, it may not be suitable for comparing different HSIs (Supplementary Fig. 4c–f). Although similarity scores for Raman spectra in the CH-stretching region are typically close to 1, the variance within pixels of a pure sample is much lower (Supplementary Fig. 5). These results show that SRS HSI is suitable for PRM analysis.

## Mapping cholesterol levels in *Drosophila* fat body using reference spectra

As a proof of concept, we applied the PRM algorithm to detecting and comparing cholesterol levels in fat body tissues of young and old *Drosophila*. Analogous to mammalian liver and adipose tissue, *Drosophila* fat body has been used extensively to study lipid metabolism. We collected fat body tissue spectra from young and old flies using spontaneous Raman spectroscopy and compared them to reference spectra of cholesterol at the fingerprint (750–1650 cm$^{-1}$) and CH-stretching (2700–3150 cm$^{-1}$) regions (Fig. 3a–d). Although the fat body is known to be enriched in triacylglycerides (TAGs), PRM enabled us to

extract cholesterol-matched signals in a TAG-rich environment using the cholesterol reference standard. Compared to samples from young flies, fat body samples from old flies showed significantly higher similarity scores to the cholesterol reference spectra in both fingerprint and CH stretching regions (Fig. 3e, f), indicating elevated cholesterol content in old flies. This result is consistent with the published data[14]. This analysis demonstrates our PRM algorithm as an effective method for rapid in situ lipid mapping in tissues.

Depending on the biological questions to address and Raman scattering equipment available, either the CH-stretching or fingerprint region in a Raman spectrum may be the focus of a study. Both regions can be used to analyze changes in biomolecule distribution, pathological structures (such as amyloid plaques), and other morphological characteristics[15–19]. Although both spectral regions yielded similar results, the fingerprint region generated results with a lower rejection level of $p < 0.001$ (Fig. 3). This is likely because the fingerprint region contained more definitive features, and the CH-stretching region possessed low intensity shoulders below 2800 cm$^{-1}$ and above 3000 cm$^{-1}$, which may lead to a higher similarity score between samples since both spectral data sets matched in those regions where the intensity was zero. Importantly, this demonstrates that similarity scores generated from spectra data by PRM-SRS can be used to estimate the levels of biomolecules in the samples.

## Using PRM-SRS to detect cardiolipin changes in cells

After validating the efficacy and robustness of PRM on spontaneous Raman spectral analysis, we next extended the algorithm to the analysis of stimulated Raman scattering (SRS) images. To evaluate the spatial accuracy and quantitative approximation of PRM-SRS, we first benchmarked it against fluorescence microscopy images. Using PRM-SRS imaging, we examined cardiolipin (CL), an essential phospholipid in the inner mitochondrial membrane, in cultured HEK293 cells. CL is synthesized in the inner mitochondrial membrane in consecutive reactions catalyzed by enzymes including phosphatidylglycerophosphate synthase 1 (PGS1), phosphatidylglycerophosphate phosphatase (PTPMT1) and cardiolipin synthase (CLS1)[20,21]. PGS1 is essential for CL synthesis, and expression of an enzyme-deficient mutant PGS1 leads to a reduction of PGP (Phosphatidylglycerophosphate) and CL in CHO cells[22]. We generated stable HEK293 cell lines with downregulated PGS1 (shPGS1). PGS1 downregulation was confirmed by immunofluorescence analysis using a PGS1-specific antibody (Supplementary Fig. 6). Following staining using nonyl acridine orange (NAO), a fluorescent dye with high affinity for CL[23], cells were analyzed using both two-photon fluorescence (TPF) microscopy and PRM-SRS. To demonstrate the specificity of SRS signals for CL, we compared control HEK293 cells with shPGS1 cells. PRM-SRS analysis of the hyperspectral images was consistent with TPF images in both control and shPGS1 cells (Fig. 4a, b). Quantitative analyses of both PRM-SRS images and fluorescence images showed significant decreases of CL signals in shPGS1 cells compared with control cells (Fig. 4c, d). Importantly, the similarity score image of the reference-matched CL is distinct from any single Raman shift images in the CH symmetric stretching regions (Supplementary Fig. 6c). These results demonstrate the ability of PRM-SRS to quantitatively detect CL changes in cells, and its potential for visualizing lipid metabolic dynamics at the subcellular scale.

To compare the image similarity between fluorescence image and PRM-SRS image, the similarity index and normalized mean squared error were calculated (Supplementary Fig. 7). We found that the similarity index between fluorescence images of NAO stained cardiolipin and PRM-SRS images of cardiolipin was higher than other lipid subtypes. Normalized mean squared error for cardiolipin is lower than that of other lipid subtypes, supporting the highest similarity between NAO stained cardiolipin and PRM-SRS image of cardiolipin. Importantly, down-regulating PGS1, an enzyme critical for cardiolipin biosynthesis, significantly reduced NAO-staining signal and PRM-SRS

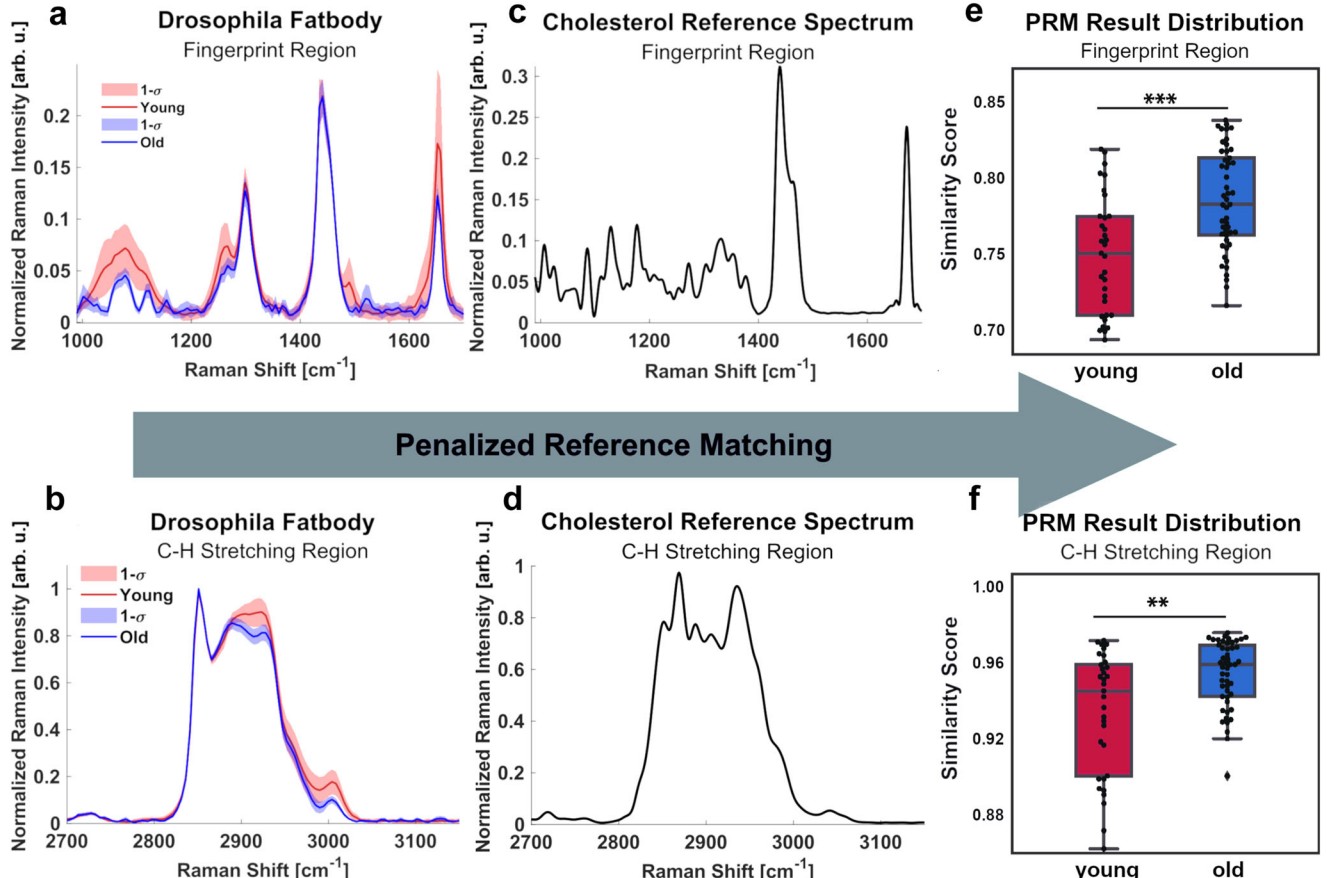

**Fig. 3 | Spectral PRM cross correlation in the fingerprint and CH stretching regions. a, b** Raman signals from fat body tissues of young ($n = 33$) and old ($n = 50$) biologically independent *Drosophila* in the fingerprint and CH stretching regions. Error bands represent 1 standard deviation (SD) from the mean. **c, d** Raman signals of the cholesterol reference standard in fingerprint and CH stretching regions, respectively. **e** Fingerprint region similarity scores of young ($n = 33$) and old ($n = 50$) *Drosophila* fat body samples to the cholesterol reference standard. Box plots indicate median and interquartile range, with whiskers indicating ±1.5 times the interquartile range. $p = 4.85 \times 10^{-4}$ by two sided Wilcoxon rank sum test. **f** CH-stretching region similarity scores of young and old *Drosophila* fat body samples to the cholesterol reference standard. Box plots indicate median and interquartile range, with whiskers indicating ±1.5 times the interquartile range. $p = 0.0037$ by two sided Wilcoxon rank sum test. \*\*$p < 0.01$; \*\*\*$p < 0.001$. PRM penalized reference matching. Source data are provided as a Source Data file.

measured cardiolipin signal, supporting that PRM-SRS measured cardiolipin signals reflect the cardiolipin levels in the samples. These results of similarity comparison show that PRM-SRS describes well the cardiolipin distribution.

## PRM-SRS tracking clinically relevant lipid subtype biomarkers in human kidney tissue

We then applied PRM-SRS to characterizing lipid subtypes in human kidney tissues, a structurally and functionally highly complex tissue composed of more than 50 cell types[24]. Cholesterol, ceramides (Cer), and triacylglycerides are among the most abundant lipid species in the kidney. Dyslipidemia is frequently observed in nephrotic syndrome (NS) and various types of chronic kidney disease (CKD)[25]. The glomerulus, the filtration unit of the nephron, is a network of capillaries that sequesters lipid species as an initial step of filtration and is decorated with lipid droplets. Wrapping around the capillary of the glomerular tuft are podocytes, making up the epithelial lining of the Bowman's capsule. We used healthy human kidney tissue sections as a control to showcase the application of our PRM-SRS in imaging different lipid subtypes in structurally complex tissue samples.

SRS imaging detected the overall distribution of lipids in the morphologically distinct structures in the kidney tissue, such as glomeruli, tubules, and blood vessels (Fig. 5a). Using PRM-SRS, we estimated relative concentrations of lipids in different structures, such as lipid droplets in podocytes, and eosinophilic bodies near tubules

(Fig. 5a, b). PRM-SRS imaging revealed distributions of distinct lipid subtypes in the glomerulus and surrounding structures in situ, including TAG, cholesterol, cholesterol ester, and C12 ceramide, with the 90th percentile similarity scores to the corresponding pure lipid reference spectra (Fig. 5c).

Dyslipidemia is manifested as elevated levels of serum TAGs, cholesterol, and very low to intermediate density lipoproteins. Common initial abnormalities include decreased production and activity of lecithin-cholesterol acyltransferase which decreases high-density lipoprotein (HDL) levels and maturation of HDL cholesterol[26]. The regulation of HDL cholesterol is tightly controlled by several organs, but generally entails the esterification of cholesterol into cholesterol esters, which move towards the center of HDL particles, along with neutral TAGs. This maintains a favorable cholesterol gradient as these HDL particles become enriched by sequestering cholesterols and fatty acids from other lipoproteins. Although mature lipoproteins are too large to pass the glomerular filtration barrier, lipids and lipid-bound proteins from lipoproteins may affect overall renal lipid metabolism[27]. Our ratiometric imaging revealed that there is a greater amount of non-esterified cholesterol in the lipid particles than neighboring structures. These pools of cholesterol may represent those yet to be enriched or ectopic deposits (Supplementary Fig. 8). Ceramides are also abundant in the kidney and play a crucial role in regulating cellular processes and binding cholesterol and other lipoproteins[28]. Ceramides, e.g., C12 ceramide, show high similarity with pixel spectra in

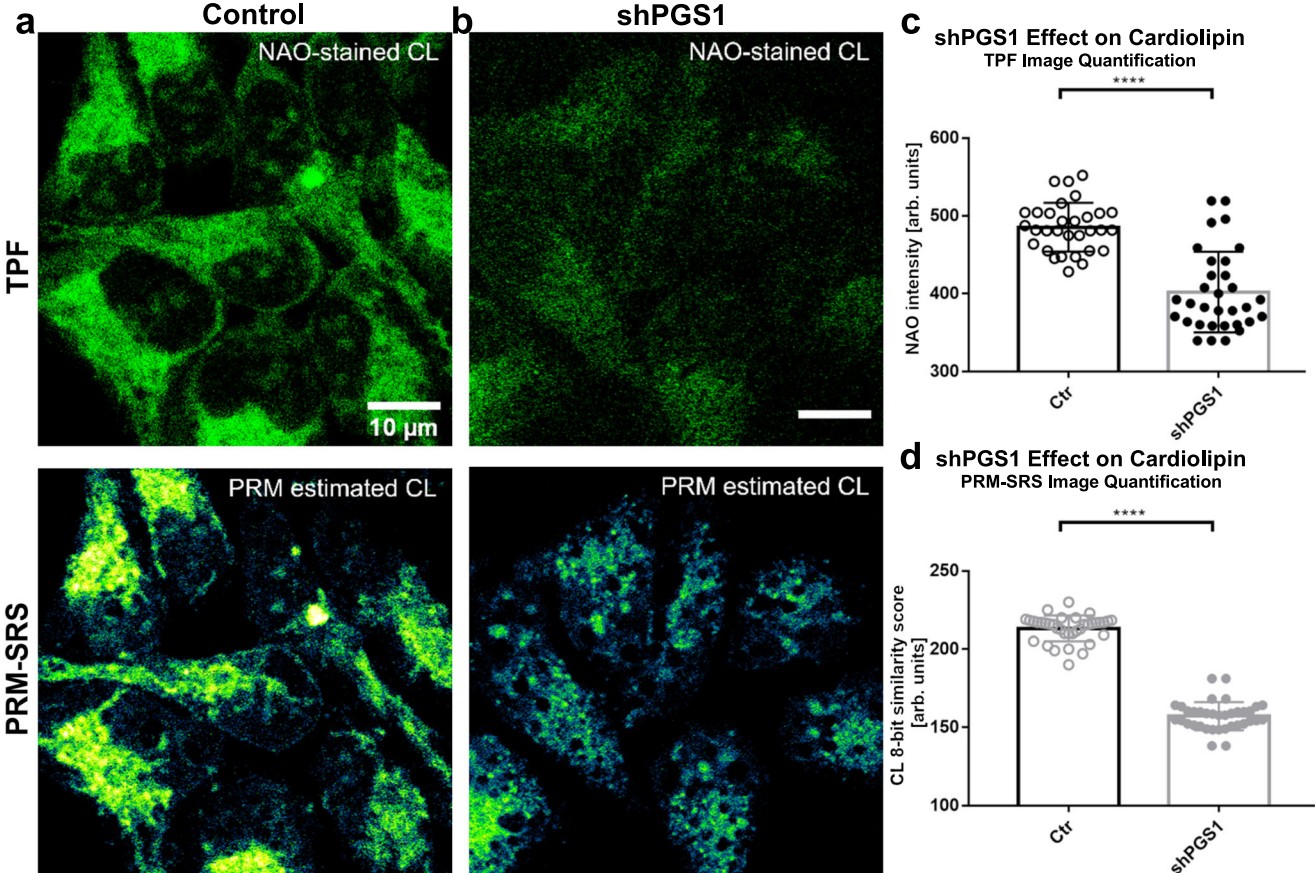

**Fig. 4 | PRM-SRS and fluorescence staining show similar results.**
**a**, **b** Comparison of PRM-SRS and fluorescence staining in control cells expressing shCtr (Ctr; **a**) and PGS1 knockdown (shPGS1; **b**) HEK293 cells. Panels on the top, two photon fluorescence microscopy (TPF) images following nonyl acridine orange (NAO)-labeling of CL. Panels on the bottom, label-free SRS hyperspectral images of CL at the CH-stretching region. **c**, **d** Quantitative analyses of NAO staining signal intensity and PRM-SRS imaging signal intensity (presented as similarity score in an 8-bit image) of CL in control ($n = 31$ cells over 2 technical replicates) and PGS1 knockdown (shPGS1; $n = 32$ cells over 2 technical replicates) cells. Significantly decreased signals in shPGS1 cells were detected by both TPF and PRM-SRS microscopy. Values are mean ± SEM. ****$p < 0.0001$ by Student's *t*-test. $n = 3$ biologically independent experiments. Scale bar, 10 μm. TPF two photon fluorescence, NAO Nonyl-acridine Orange, CL Cardiolipin, shPGS1 short hairpin phosphatidylglycerophosphate synthase 1. Source data are provided as a Source Data file.

lipid droplets and lipoprotein particles (Fig. 5c). In nephropathies, ectopic lipid deposits in the glomerular mesangium and proximal tubules are typically concurrent with low HDL levels[26]. The ability of PRM-SRS to track the lipidomic profile in tissues collected from patients at various stages of diseases will generate critical data for changes in these macromolecules over time, and with associated biological variables. Such studies will provide insights into assessing severity, progression or prognosis of various lipid metabolic diseases.

**Mapping lipid subtype distributions in *Drosophila* fat body**
Using maximum intensity projection (MIP) of the PRM-SRS hyperspectral image of total lipids, we visualized lipid droplets in *Drosophila* fat body cells (Fig. 6a). We also detected lipid subtypes using different lipid reference standards, including TAG and phosphatidylethanolamine (PE). In addition to detecting lipid subtypes, PRM-SRS can also provide information on subcellular distribution, including co-localization, of different lipid subtypes. Comparison of MIP of the PRM-SRS hyperspectral image of total lipids (Fig. 6a) with mono-unsaturated triacylglycerol (TAG 18:1) reference-matched image (Fig. 6b) revealed abundant TAG in lipid droplets (Fig. 6a, b). A critical tenet of unmixing techniques such as PRM is that spectral shape, rather than intensity, drives the similarity score of normalized spectra. The MIP in Fig. 6a shows several lipid droplets with non-uniform maximum intensities, yet the TAG reference matched image shows a more uniform result. This was because

despite intensity differences that may have arisen from the sample focus plane, the spectral shapes were still consistent.

*Drosophila* fat body cells contain lysosome-like structures that regulate their lipid anabolism and these structures were detected using reference spectra (Fig. 6c). These spectra, unlike those of individual lipid subtypes, have a more dominant $CH_3$ stretching peak at 2935 cm$^{-1}$, and a more pronounced olefinic peak at 3065 cm$^{-1}$. PE is one of the most prominent lipid subtypes in cell/organelle membranes and can be visualized by taking PRM-SRS images using their corresponding reference standard. Figure 6d shows the spatial distribution of the ratio of the PE and TAG similarity-scored images. The merged image of the aforementioned lipid subtypes is shown in Fig. 6e. Upon closer inspection, Fig. 6f shows the spectra of the apparent pixels are similar to TAG, and therefore appear darker in those regions in both the TAG and PE similarity-scored images. However, the lipid cores have a greater disparity in these similarity scores, with an even greater similarity to TAG and lesser similarity to PE. Therefore, the apparent pixels are visible because of relative concentrations, as discussed in Supplementary Fig. 2. Reference spectra for the respective lipid subtypes are overlaid with the mean spectrum from the 90th percentile pixel similarity scores (Fig. 6g). The intensity profiles (Fig. 6h) support the notion that signal intensity of images based on similarity scores varies by spectral shape, whereas signal intensity in SRS images varies by chemical bond concentrations. Thus, a lipid droplet core may appear more uniform in a TAG reference-matched PRM-SRS image than in an

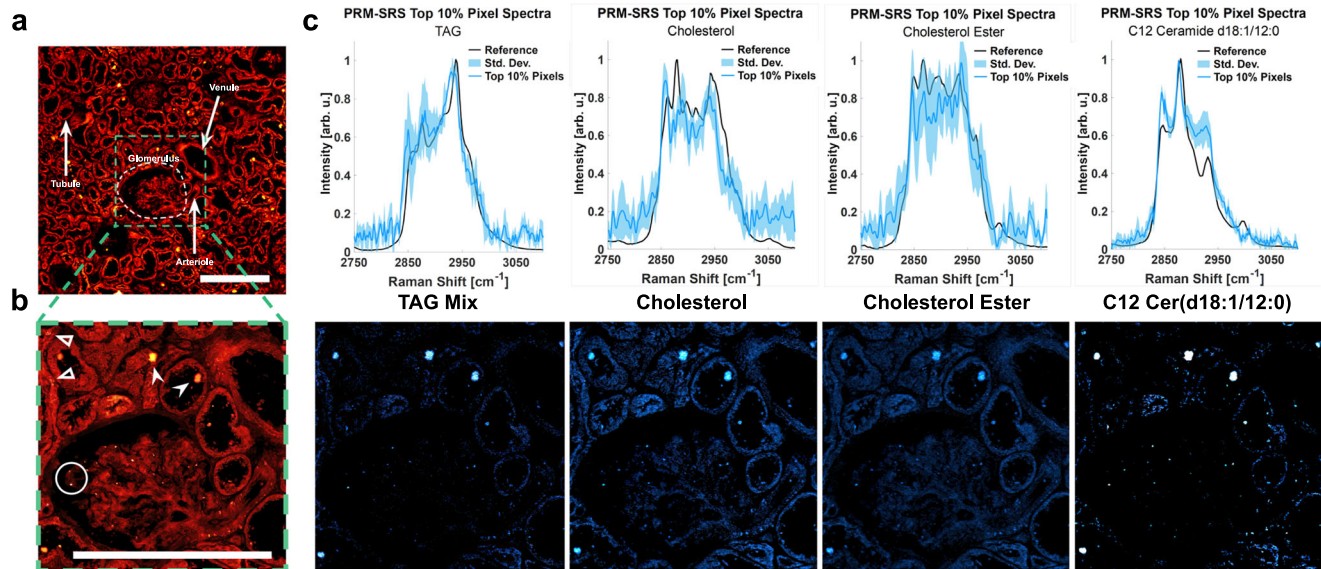

**Fig. 5 | Label-free hyperspectral detection of different lipid subtypes in situ using PRM-SRS. a**, **b** SRS image of a human kidney (*n* = 1, no replicate trials) tissue section at 2850 cm⁻¹. Panel **b** shows the enlarged image of the boxed area in panel (**a**). Hollow arrowheads, intracellular lipid droplets in tubules. Solid arrowheads, eosinophilic bodies. Circles, lipid droplets sequestered by podocytes in the glomerulus. Scale bar, 200 μm. **c** PRM-SRS spectra and images of different lipid subtypes of interest show the distribution of the similarity scores, each with the same contrast levels. Spectra of the top 10% of similarity score pixels overlaid on the reference spectrum for each lipid subtype show consistent matches. Resulting similarity score images were background subtracted to improve the contrast. Center-line of spectra correspond to mean spectrum for each respective lipid subtype with 1 standard deviation (SD) error bands. Scale bar, 200 μm. TAG triacylglycerides, Cer ceramide. Source data are provided as a Source Data file.

original SRS image of the CH₂ stretching region. Together, these data demonstrate that PRM-SRS is useful in detecting different lipid subtypes and their distribution at the subcellular scale.

### Analyzing lipid subtypes in mouse brain samples

We next applied PRM-SRS to analyzing lipid metabolism in the context of the aging using mouse hippocampal samples. We visualized and compared cholesterol, PC, and PE levels in hippocampal samples from young (3 months) and old (18 months) mice (Fig. 7a–c; f, g). We also generated ratiometric images for quantitative analysis, since the signal intensity has a linear relationship with the concentration of chemical bonds of the molecules detected. Ratiometric imaging analyses showed increased Cholesterol/PE ratio in subregions of granule cell nuclei (Fig. 7d, i; red circles). This increase in the Cholesterol/PE ratio was more prominent and detected in more granule cells in the old brain samples as compared with the young ones (compare Fig. 7d with 7i), indicative of altered cholesterol and/or PE metabolism in the old brains. These results show that ratiometric PRM-SRS imaging can detect changes in differential spatial distribution of various lipid subtypes even when such changes are not obvious in images of individual lipid subtypes.

Ratiometric images of PC/PE showed higher levels of PC relative to PE in the granule cell nuclei of the dentate gyrus in both young and old mice, but lower levels outside the nuclear regions (Fig. 7e, j). Compared to the young brain sample, the old brain sample showed no significant changes in the average PC or PE levels in the granule cells in both individual imaging channels (Fig. 7b, c, g, h) and ratiometric images (Fig. 7e, j). This is consistent with the results from Gas Chromatography Mass Spectrometry (GC-MS) (Fig. 7k, l). However, we noticed spatial distribution differences in the PC to PE ratio between young and old samples. The ratiometric images reveal that more granule cell nuclei had uniformly higher PC/PE ratio in the old brain sample, whereas the nuclei in the young sample showed less even distribution of the PC/PE ratio (red areas; see those nuclei marked by purple arrows) (Fig. 7e, j). These data suggest altered synthesis, accumulation or clearance of PC and/or PE in the granule

cells in the old brains, consistent with a previous report[29]. Since PE is a precursor of PC, higher PC to PE ratios inside the older hippocampal granule cells suggest that aging brains may have altered CTP:phosphocholinecytidylyl transferase (CCT) activity−a rate limiting PC synthesis enzyme with a predominantly nuclear localization[30]. This finding is significant because both PRM-SRS imaging and GC-MS analysis show that the young brain samples contain less cholesterol relative to PE than old ones. However, only through ratiometric analysis were we able to detect differential subcellular distribution of lipids, including cholesterol/PE and PC/PE ratio in the nuclei (Fig. 7d, i, e, j).

For comparison, we analyzed the same samples using GC-MS to quantify cholesterol/PE and PC/PE ratios (Fig. 7k), which demonstrated an increased cholesterol/PE ratio and no changes in the PC/PE ratio in old brain samples compared with young ones. Additional results from other internal lipid standards can be found in the supplementary Fig. 9. The PRM-SRS images of nuclei in the tissues were manually segmented using ImageJ for quantification of cholesterol/PE and PC/PE ratios (Fig. 7l). Shotgun lipidomics indicate several PC and PE subspecies are significantly upregulated or downregulated, as shown in the volcano plot (Fig. 7m). These data suggest that PRM-SRS may be used for quantitative lipidomic imaging analyses in tissue samples in the future.

### Detecting lipid subtype distributions in human brain tissues

Sphingosine is another crucial lipid subtype whose metabolic alteration has been suggested as a biomarker for neurodegenerative diseases, such as Alzheimer's, Parkinson's, and Huntington's diseases[4,31]. To visualize individual cells, we used label-free optical SRS histology (SRH) imaging of human brain sample (See supplementary Fig. 10) to create virtual histology images similar to that of hematoxylin-and-eosin (H&E) staining as previously reported[32]. Using PRM-SRS, we visualized sphingosine and CL simultaneously in the human brain tissue sections (Fig. 8a, b). Superimposition of sphingosine and CL images illustrates their relative distribution in brain cells (Fig. 8c, d). Ratiometric imaging (Fig. 8e) and quantitative analyses (Fig. 8f) demonstrated a clear reduction in the CL to sphingosine ratio inside

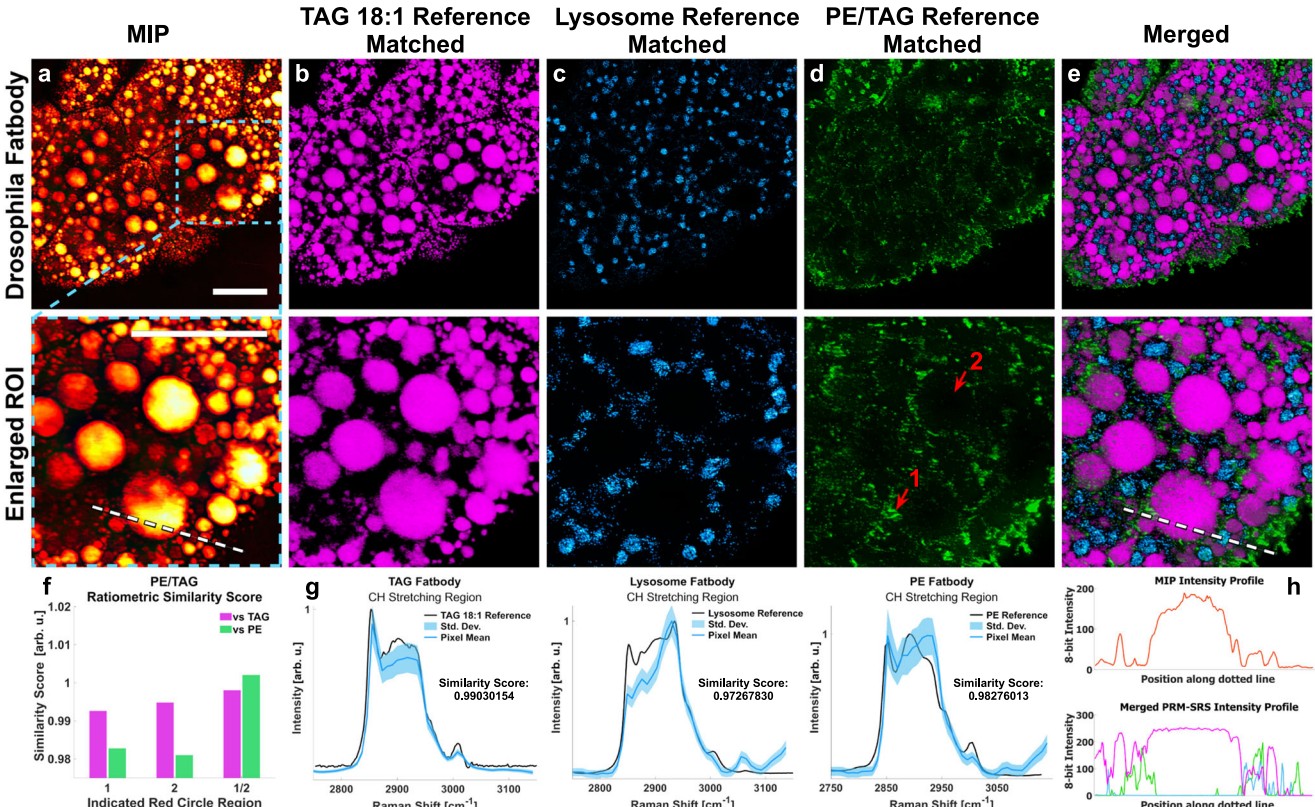

**Fig. 6 | PRM-SRS imaging of *Drosophila* fat body cells detects different lipid subtypes and their subcellular distributions. a** Maximum intensity projection (MIP) of the PRM-SRS hyperspectral image of total lipids reveals lipid droplets. **b** PRM-SRS detected TAG in lipid droplet cores. **c** Lysosome-like structures detected by PRM-SRS using reference spectra measured from lysosome-like structures in *Drosophila* fat body. **d** PE:TAG ratiometric images show that the interstitium between lipid droplet cores and lysosome-like structures has higher relative levels of PE. **e** PRM-SRS subtype images are merged to detect co-localization of different lipid subtypes. **f** Similarity scores from areas marked by red arrows 1 and 2 in the lower part of panel D highlight the necessity of evaluating relative concentrations as opposed to absolute concentrations. **g** SRS spectra of top 90th percentile pixel similarity scores in (**b**–**d**) lipid subtypes with respective standard reference spectrum overlaid. Center-line of spectra represent mean with 1 standard deviation (SD) error bands. **h** Intensity profiles along the dotted white lines in (**a**) and (**e**), upper and lower panels respectively, show how signal intensity varies with spectral shape, rather than concentration in a PRM-SRS image. *n* = 2 replicate trials. Scale bar, 20 μm. MIP maximum intensity projection, TAG triacylglyceride, PE phosphatidyl-ethanolamine. Source data are provided as a Source Data file.

the nucleus, consistent with the fact that CL is mainly localized at the inner mitochondrial membrane but not in the nucleus. These results show that PRM-SRS can be used to visualize the subcellular distribution of different lipid subtypes.

## Discussion

In this study, we developed a PRM algorithm that can efficiently unmix and distinguish a variety of lipid subtypes from single SRS HSI stacks. Compared with fluorescence imaging, our PRM-SRS platform shows the advantages of multiplexed lipid subtype visualization from single label-free HSI sets. This also represents a significant expansion in applications compared with traditional SRS imaging, which often relies on detecting total lipids in the CH-stretching mode at 2850 cm⁻¹.

With an improved contrast, PRM-SRS imaging enables us to identify different lipid subtypes. The spectra of lipid subtype standards collected from spontaneous Raman scattering microscopy can be utilized in analyzing HSI data collected from SRS microscopy. Our PRM-SRS can generate both co-localization and ratiometric data of individual lipid subtypes simultaneously by mapping their spatial distributions and quantifying their relative concentrations. In this study, we established a Raman spectra library with 38 lipid subtype standards (Supplementary Table 1) and demonstrated the simultaneous detection of a few selected lipid subtypes by PRM-SRS in cells and tissues (Figs. 3–8, Supplementary Fig. 8). Analyses of human kidney tissue samples indicate that PRM-SRS can be used to identify different lipid

subtypes associated with renal diseases, suggesting potential application of PRM-SRS in diagnosis and prognosis of these diseases, including those associated with dyslipidemia. Such label-free methods may be instrumental in the early detection of kidney diseases by detecting and measuring relative levels of different lipid biomarkers without the need to stain biopsied samples or perform destructive imaging, especially on limited clinical samples. Analyses of *Drosophila* fat body samples show that PRM-SRS can be used effectively in mapping spatial distributions of lipid subtypes at the subcellular scale. These results highlight the ability of PRM-SRS to selectively visualize multiple lipid subtypes in a single image with ease and freedom akin to individual-subtype labeled imaging without the need to actually label them. Analyses of mouse and human brain tissues demonstrate the importance of measuring relative lipid concentrations through ratiometric imaging, which reveals regionally different concentrations of lipid subtypes that may not be readily apparent in single-channel images. Although lipid subtypes are not measured in absolute concentrations, their relative levels are consistent with results from other modalities such as mass spectrometry (MS).

The brain is a lipid-rich organ. Lipid subtypes such as cholesterol and sphingolipids are important components of the brain. Alteration in lipid subcellular distribution and metabolism impact on brain cell function have been associated with neurological diseases. Our analyses of mouse and human brain tissues illustrate the capability of PRM-SRS in quantitatively mapping and analyzing distribution of different lipid

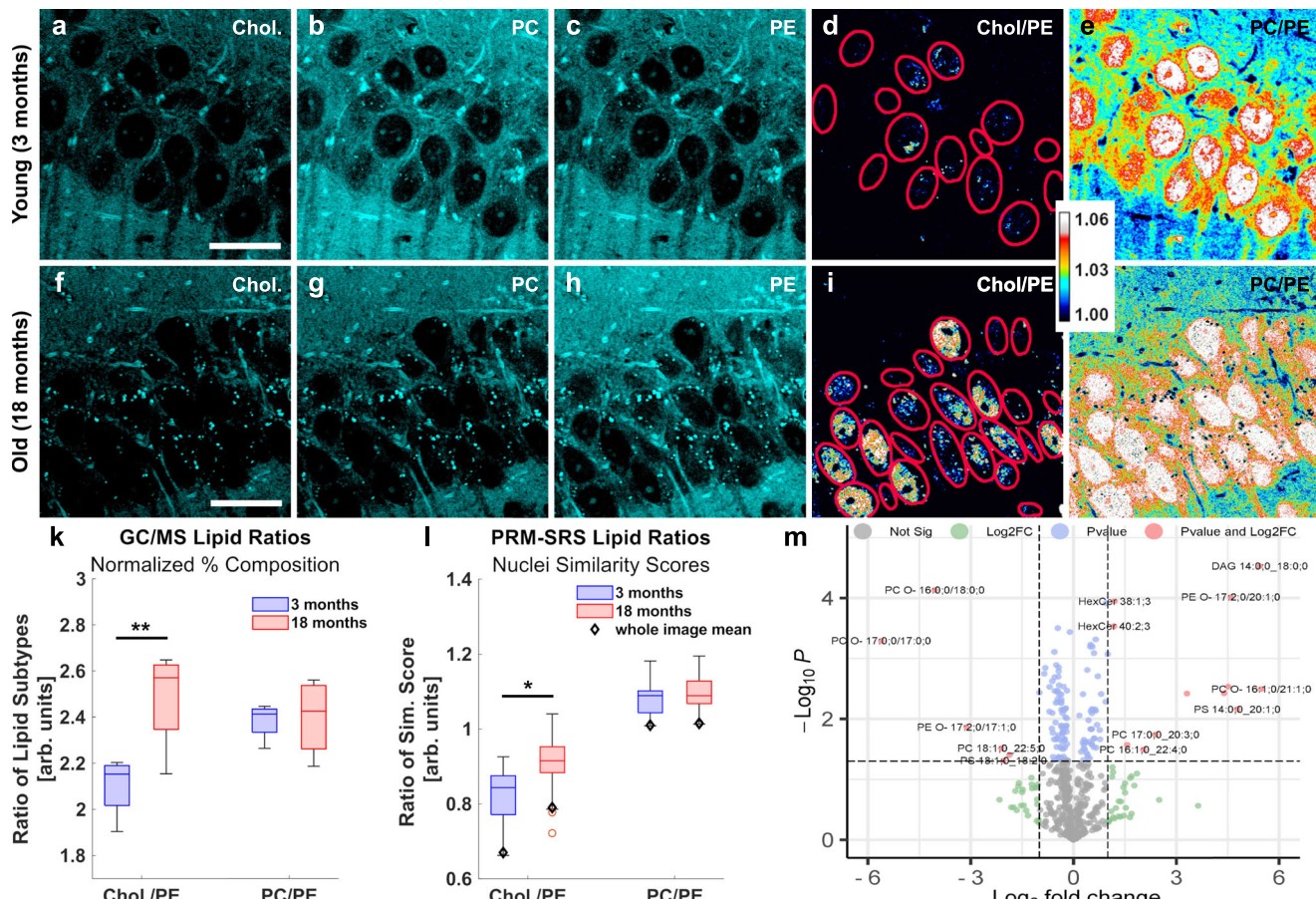

**Fig. 7 | PRM-SRS imaging of mouse hippocampal samples. a–j** PRM-SRS hyperspectral detection of cholesterol, PC, and PE in hippocampus samples from young and old mice. Overall intensity of detected lipid subtypes shows distinct patterns, with old brains showing higher cholesterol to PE ratio, but relatively consistent levels of PE and PC. **d, i** Ratiometric images of cholesterol to PE shows more nuclei with higher cholesterol/PE ratio in the old brains. Selected nuclei are marked by red circles. **e, j** Ratiometric images of PC relative to PE show higher PC/PE ratio in granule cell nuclei of both young and old brains, but the spatial distribution of the ratio is more heterogeneous in young samples (see nuclei marked by purple arrows). **k, l** Mass spectrometry (**k**) shows results consistent with that obtained by ratiometric PRM-SRS imaging (**l**) for biologically independent mice ($n = 4$) in each age group. Box plots indicate median and interquartile range, with whiskers indicating 1.5 times the interquartile range. Ratiometric image intensities, corresponding to the ratio of PRM similarity scores of lipid subtypes, are plotted. Error bars represent standard deviation (SD). *$p = 0.017$; **$p = 0.006$. **m** Volcano plot showing significantly altered lipid species ($n = 534$) between groups. $p = 0.05$, fold change > 2. Data are filtered such that only lipids present at least three times across all samples are shown. Scale bar, 20 μm. Chol cholesterol, PC phosphatidylcholine, PE phosphatidylethanolamine, GC/MS Gas Chromatography Mass Spectrometry. Source data are provided as a Source Data file.

subtypes within single cells. These analyses confirm the cross-applicability of the fingerprint and CH-stretching spectral regions for quantitative analyses. Further, our PRM-SRS imaging shows that sphingosine, a catabolic product of sphingomyelin, has a predominantly nuclear localization. Nuclear sphingomyelinase and sphingosine kinases regulate the release of ceramides and sphingosine, as well as the conversion of sphingosine to sphingosine-1-phosphate. These processes regulate cell proliferation and cell death[33]. Sphingosine kinases may shift from a cytosolic to a nuclear localization in brain samples from Alzheimer's disease patients[34]. The development of new technologies in imaging distinct lipid subtypes and their metabolism will enhance our ability to investigate molecular mechanisms underlying different brain disorders.

As shown in Fig. 4, Fig. 8, Supplementary Figs. 2 and 3, PRM-SRS has sufficient capability to provide quantitative information on lipid subtype distribution. Theoretically, the similarity scores and concentrations of dominant molecules have a linear relationship in a certain dynamic concentration range. However, spectral shifts caused by multiple interfering factors (such as change in chemical environment or instrumentation) may distort the relationship between the similarity scores and concentrations. Since the spectral shifts can be

caused by multiple factors, it is difficult to define an exact function of a spectral shift. Nevertheless, such spectral shifts can be corrected by using the penalized regression analysis. In brief, the penalty term helps us calculate the similarity score to more precisely describe the linear relationship with concentration. Depending on the equipment used and the samples of interest, careful tuning of the penalty term in the PRM algorithm is necessary. At present, the PRM-SRS platform should be used in well-controlled experiments to limit external chemometric dimensions. In this way, spectral signals are more likely from molecular subtypes in the samples, rather than from noises. Since Raman peak intensities are multiplexed in the sense that a specific peak shape may be influenced by multiple molecules, it is critical that molecular makeup is as consistent as possible when using PRM-SRS to estimate relative concentrations of different molecular subtypes. In the current PRM platform, users should determine the optimal penalty coefficient experimentally to avoid artificial increases or decreases of similarity scores. An extremely low penalty coefficient would allow the comparison between reference spectra and pixel spectra to occur at any offset, which may inflate the overall similarity score. For example, ceramide has a notably high peak at 2880 cm$^{-1}$, while pixel spectra typically have the most prominent peak at the 2940 cm$^{-1}$ area. Allowing

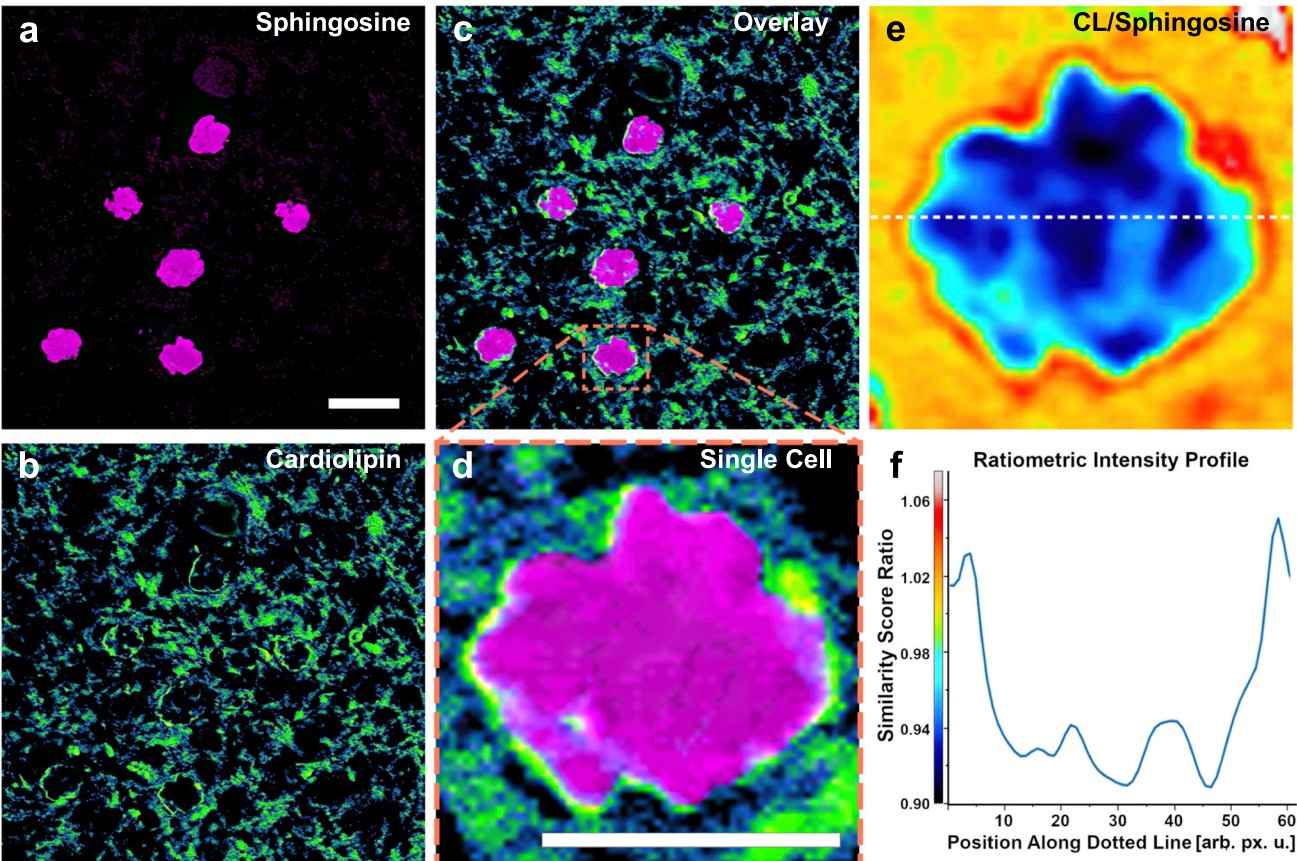

**Fig. 8 | Hyperspectral SRS imaging detection of cardiolipin and sphingosine in a human brain tissue section. a** Sphingosine similarity score image of human brain tissue. Scale bar, 10 μm. **b** CL in the same region of interest. **c** Merged image of CL and Sphingosine. $n = 3$ technical replicates for (**a–c**). **d** Zoomed-in image of a single brain cell with CL and Sphingosine similarity scores. Scale bar, 5 μm. **e** Ratiometric image of CL to Sphingosine similarity scores in the single brain cell in (**d**). **f** Intensity profile of (**e**) along the indicated white dashed line. CL cardiolipin. Source data are provided as a Source Data file.

spectral offsets without penalty may result in a high similarity score because the ceramide reference could yield a high similarity score with the pixels with a sharp 2940 cm$^{-1}$ peak. On the other hand, an extremely high penalty coefficient would be akin to not allowing spectral offsets during comparison at all, which would be similar to traditional reference matching. This is disadvantageous because spectra acquired with different equipment may not be exactly calibrated on the same $x$ axis, which could artificially decrease the similarity scores. While these cases do not occur often, the greatest similarity score does occur at very small offsets. Figure 2d–f shows PRM-SRS images with different penalty coefficients. Although the PRM-SRS pipeline can be further enhanced by including more comprehensive reference standards and further increasing analysis speed, this platform is robust for analyzing different lipid subtypes. Reference matching could also be a useful tool to detect the presence of representative mixtures of compounds, not simply individual molecules.

The main advantages of PRM-SRS include multiplexed molecular subtype visualization, positive values, and fast speed of similarity score calculation. Similarity scores are always positive values since Raman intensities cannot be negative. On the other hand, pseudo-inverse matrix (PINV) coefficients can be negative, which will make it difficult to normalize the output. Our similarity score calculation is faster than other methods, such as the pseudo-inverse matrix (Supplementary Fig. 3). PRM and PINV show similar results in correlation with relative concentrations. However, PINV-based calculation time increases exponentially with the number of spectra in the original matrix and image size. When analyzing 1024 × 1024 hyperspectral images using PINV, there are millions of spectra in a single experiment. On the other hand, similarity score calculation using PRM-SRS is based on the inner-product, which is easily vectorized and split in a parallel pool. The calculation time difference between PRM and PINV is shown in Supplementary Fig. 2. Considering the number of spectra in a single HSI stack, the calculation speed is an important factor as a practical analysis tool for image analysis. Although the PRM-SRS result cannot provide absolute concentrations as precisely calibrated linear unmixing methods, this approach clearly shows advantages over other methods.

PRM-SRS can also be used together with other chemometric methods, such as GC-MS for cross validation, as the incidence of false positive may still be high. Finally, detection of the vast variety of lipid subtypes may require further improvement in unmixing methods and spectral resolution, as the lipid subtype reference library is expanded, and more lipid subtypes are further evaluated. With some adjustments, such as using different reference libraries, this PRM-SRS platform can be extended to analyzing other molecules, including proteins, nucleic acids, and even clinically relevant molecular complexes (such as protein aggregates or oligomers).Using heavy water (D$_2$O) probed SRS (DO-SRS), metabolic imaging can also distinguish de novo newly synthesized biomolecules, including lipids, proteins, and DNA[32,35,36], from old existing biomolecules in cells and tissues at the subcellular resolution. This ever-expanding library of molecular subtype references may warrant broader spectral regions, including the fingerprint, CH-stretching, and O-H stretching regions to increase the chemometric dimensions. In this study, due to the stronger signal in the CH-stretching region than the fingerprint region, we focused on the CH-stretching region

to analyze lipids. In our future study applying PRM-SRS to visualize protein distribution, the fingerprint region will be a focus of our attention. To achieve this goal, integration of statistical denoising and regression methods will help increase the power of molecular subtype matching. Application of higher order signal manipulations such as digital derivatives and wavelet analyses, will enhance the ability to extract the most prominent as well as subtle but important features.

In summary, this study presents a hyperspectral imaging platform—PRM-SRS—that allows for direct identification of multiple molecular species in situ with subcellular resolution and high chemical specificity by leveraging the cross-applicability of spectral reference libraries and HSI methods. This PRM-based method can be applied to various microscopy setups, such as SRS, FTIR, and spontaneous Raman scattering spectroscopy. Compared with existing HSI methods, PRM-SRS shows a much enhanced speed and efficiency. With appropriate reference spectra established, PRM-SRS can be used to detect a wide range of different biomolecules. This platform can also be applied to studying metabolism of diverse types of biomolecules in cell and tissue samples. For example, when combined with transcriptomics analyses following up- or down-regulation of lipid metabolism genes, it will be highly useful for investigating metabolic changes under different pathophysiological conditions. With its easy implementation, PRM-SRS can be combined with high-throughput methods, such as microfluidic/nanofluidic devices and single-cell apparatuses, or with large-area HSI mapping methods. The application of deep learning algorithms, such as DeepChem, may further improve the imaging speed in femtosecond SRS imaging[10]. PRM-SRS could benefit from additional instrumentation improvements, such as distortion-free polygon scanning and spectral focusing, as well as from machine learning to enhance the SNR[37]. Finally, PRM could easily augment other unmixing methods, including MCR-ALS, by providing fast initial component spectra. Thus, PRM-SRS has great potentials in multiplex cell and tissue imaging with a broad spectrum of applications.

## Methods
### Ethical statement
Applicable experimental protocols were approved by the Institutional Animal Care and Use Committee (IACUC) at the University of California San Diego and the IACUC at Duke University, respectively. The approved IRB in WashU St. Louis has signed off on the MTA (17833) Transfer Agreement with UC San Diego to ensure the sharing is consistent with the approved IRB application.

### Sample preparation
**HEK293 cell cultures.** The parental HEK293 cell line was obtained from the American Type Culture Collection (ATCC). Cells were cultured in DMEM supplemented with 5% fetal bovine serum (FBS), 1% penicillin/streptomycin (Fisher Scientific, Waltham, MA).

The control shRNA construct was as previously described[38]. A shPGS1 construct was designed PGS1 by expressing shRNA against PGS1 (target sequence: 5'-TCGGGTTCCATCCGTTTAAAT-3') in the plasmid vector Tet-pLKO-puro (Vector Builder Inc.) to specifically down-regulate expression of PGS1. The control and shPGS1 constructs were transfected into HEK293 cells using lipofectamine (Invitrogen). Following transfection, cells were selected using puromycin (1 μg/ml) and stably expressing cell clones were obtained. Control and shPGS1 cells were passaged at 80% confluence and plated on #1 thickness laminin-coated coverglasses (GG12-laminin, VWR). After allowing cells to adhere to the coverglasses for 2 h, cells were fixed using 4% v/v PFA for 15 min and stained with 100 nM NAO in the dark for 30 min. Cells were SRS imaged transmissively through coverglasses.

Immunofluorescence staining was performed following our published protocol[38] using a polyclonal rabbit anti-PGS1(Sigma-Aldrich, Cat# AV48896) and secondary antibody conjugated with Alexa-488 (Abcam, Cat# ab150081).

**Human kidney tissue preparation.** De-identified human kidney tissue sections (30 μm) were prepared from 4% v/v PFA-fixed frozen biopsy samples using a Compresstome (VF-210-0Z, Precisionary). The kidney cortex was isolated for imaging. Samples were imaged between 1 mm thick glass slide and #1 thickness coverglass, submerged in 1× PBS.

**Human brain tissue preparation.** De-identified post-mortem autopsy human brain sections (6 μm) were prepared from formalin-fixed and paraffin-embedded cortex tissue of control subject without detectable neuropathology as previously published[39,40]. The sections were deparaffinized following a published protocol[13]. Subsequent SRS imaging was conducted with the tissue sections sandwiched in PBS between 1 mm thick glass slides and #1 thickness cover glass.

**Mouse Brain Samples.** Young (3 months) and aged (18 months) mice were euthanized with 5% isoflurane, and then perfused with 4% paraformaldehyde. The brains were harvested and fixed in 4% paraformaldehyde at 4 °C for overnight. The fixed brains were washed with PBS and cut into 120-μm thickness slices with Vibratomes (Precisionary). The brain slices were placed in the center of a spacer and sandwiched between glass slides and coverslip for hyperspectral SRS imaging.

***Drosophila* fat body samples.** Wild type (*w^1118* stock #5905) were originally obtained from the Bloomington Stock Center and have been maintained in the lab for several generations. Fat bodies were dissected from day 7 adult flies and fixed in 4% PFA (in 1×PBS) for 15 min. Samples were imaged immediately using SRS microscopy for hyperspectral imaging. Whole animal experiments were conducted in compliance with all relevant ethical regulations and based on UCSD approved protocol.

### Spontaneous Raman spectroscopy
Spontaneous Raman scattering spectra were obtained by a confocal Raman microscope (XploRA PLUS, Horiba) equipped with a 532 nm diode laser source and 1800 lines/mm grating. The acquisition time is 30 s with an accumulation of 4. The excitation power was ~40 mW after passing through a 100× objective (MPLN100X, Olympus). Output spectra were background subtracted and vector and simplex normalized. The pure lipid reference standards were placed on glass slides for spontaneous Raman spectra measurement. All lipid subtype reference spectra were acquired in the same manner.

### Stimulated Raman scattering microscopy
An upright laser-scanning microscope (DIY multiphoton, Olympus) with a 25× water objective (XLPLN, WMP2, 1.05 NA, Olympus) was applied for near-IR throughput. Synchronized pulsed pump beam (tunable 720–990 nm wavelength, 5–6 ps pulse width, and 80 MHz repetition rate) and Stokes (wavelength at 1032 nm, 6 ps pulse width, and 80 MHz repetition rate) were supplied by a picoEmerald system (Applied Physics & Electronics) and coupled into the microscope. The pump and Stokes beams were collected in transmission by a high NA oil condenser (1.4 NA). A high O.D. shortpass filter (950 nm, Thorlabs) was used that would completely block the Stokes beam and transmit the pump beam only onto a Si photodiode for detecting the stimulated Raman loss signal. The output current from the photodiode was terminated, filtered, and demodulated in X with a zero phase shift by a lock-in amplifier (HF2LI, Zurich Instruments) at 20 MHz. The demodulated signal was fed into the FV3000 software module FV-OSR (Olympus) to form the image

during laser scanning. All SRS images were obtained with a pixel dwell time 40 μs and a time constant of 30 μs. A stack of 512 pixel ×512 pixel ×76 images in the C−H stretching region took ~15 min to acquire. The PRM analysis of this image-stack took less than 1 min. Laser power incident on the sample was approximately 40 mW. Stimulated Raman histology was performed following published protocol[41].

### Gas Chromatography Mass Spectrometry (GC-MS)

Hippocampal slices ($n = 4$ per group) from 3-month-old and 18-month-old mice were homogenized in ethanol/water1:1 (v/v) and the homogenate were sent to Lipotype GmbH (Dresden, Germany) for mass spectrometry-based lipid analysis[42]. Lipids were extracted using a two-step chloroform/methanol procedure[43]. Samples were spiked with internal lipid standard mixture containing: cardiolipin 14:0/14:0/14:0/14:0 (CL), ceramide 18:1;2/17:0 (Cer), diacylglycerol 17:0/17:0 (DAG), hexosylceramide18:1;2/12:0 (HexCer), lyso-phosphatidate 17:0 (LPA), lyso-phosphatidylcholine 12:0 (LPC), lyso-phosphatidylethanolamine 17:1 (LPE), lyso-phosphatidylglycerol 17:1(LPG), lyso-phosphatidylinositol 17:1 (LPI), lyso-phosphatidylserine 17:1 (LPS), phosphatidate 17:0/17:0 (PA), phosphatidylcholine 17:0/17:0 (PC),phosphatidylethanolamine 17:0/17:0 (PE), phosphatidylglycerol 17:0/17:0 (PG), phosphatidylinositol 16:0/16:0 (PI), phosphatidylserine 17:0/17:0 (PS), cholesterolester 20:0 (CE), sphingomyelin 18:1;2/12:0;0 (SM), triacylglycerol 17:0/17:0/17:0 (TAG) and cholesterol D6 (Chol). After extraction, the organic phase was transferred to an infusion plate and dried in a speed vacuum concentrator. First step dry extract was re-suspended in 7.5 mM ammonium acetate in chloroform/methanol/propanol (1:2:4, V:V:V) and 2nd step dry extract in 33% ethanol solution of methylamine in chloroform/methanol (0.003:5:1; V:V:V). All liquid handling steps were performed using Hamilton Robotics STARlet robotic platform with the Anti Droplet Control feature for organic solvents pipetting.

Samples were analyzed by direct infusion on a QExactive mass spectrometer (ThermoScientific) equipped with a TriVersa NanoMate ion source (Advion Biosciences). Samples were analyzed in both positive and negative ion modes with a resolution of Rm/z = 200 = 280,000 for MS and Rm/z = 200 = 17,500 for MSMS experiments, in a single acquisition. MSMS was triggered by an inclusion list encompassing corresponding MS mass ranges scanned in 1 Da increments[44]. Both MS and MSMS data were combined to monitor CE, DAG and TAG ions as ammonium adducts; PC, PC O-, as acetate adducts; and CL, PA, PE, PE O-, PG, PI, and PS as deprotonated anions. MS only was used to monitor LPA, LPE, LPE O-, LPI and LPS as deprotonated anions; Cer, HexCer, SM, LPC, and LPC O- as acetate adducts and cholesterol as ammonium adduct of an acetylated derivative[45]. Data were analyzed with in-house developed lipid identification software based onLipidXplorer[46]. Data post-processing and normalization were performed using an in-house developed data management system. Only lipid identifications with a signal-to-noise ratio >5, and a signal intensity fivefold higher than in corresponding blank samples were considered for further data analysis.

### Data analysis

**Image processing.** SRS images were converted to unsigned 16 bit images via MATLAB, and were filtered using a morphological top-hat algorithm with 8 structuring elements, where appropriate. Unless used in ratiometric calculations, images for display were background subtracted using a sliding paraboloid with a radius of one tenth the image length. Intensity profiles and color maps were generated from ImageJ. All images within a figure have the same contrast unless specified otherwise. Ratiometric and overlaid images were created using the Image Calculator function and Overlay function, respectively, in ImageJ. Statistical analyses were performed using SPSS.

**Penalized Reference Matching Algorithm.** Computation was conducted as described in the main text using MATLAB R2021b using an 8 Core Intel i9-9880H CPU, NVIDIA Quadro RTX 4000, and 64GB of RAM. Spectra were intensity normalized from 0 to 1 following baseline correction using arPLS (if background spectra were not available for subtraction). All spectra were interpolated at every integer wavenumber using the interp1 function to avoid dimension mismatch errors during inner product calculations. Spectra were also Euclidean normalized using the standard vecnorm or norm functions in MATLAB. If a spectral shift of the reference spectrum exceeded the range of the original analyte spectrum, it was padded with zeros on the leading side, and trimmed on the lagging side. For timing and efficiency calculations, no parallel workers were used to split the spectral dataset for processing, but more workers are possible in MATLAB if supported by the hardware if the dataset is exceptionally large.

### Reporting summary

Further information on research design is available in the Nature Portfolio Reporting Summary linked to this article.

## Data availability

The data that support the findings of this study are provided as a Source Data File including numerical data of important plots in figures. Some of Matlab codes for simulation are also included. Source data are provided with this paper.

## Code availability

Example data and source code for PRM-SRS with explanations about parameters and installation protocol are available at https://github.com/lingyanshi2020/PRM-SRS.

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

## Acknowledgements

We thank Drs. W. Min, K. Zhang, C. Metallo, and F. Liu for helpful discussions. We acknowledge support from UCSD Startup funds, NIH R01GM149976, NIH U01AI167892, NIH 5R01NS111039, NIH R21NS125395, NIHU54DK134301, NIHU54 HL165443, NIH U54CA132378, and Hellman Fellow Award. We are grateful for the support of the Washington University Kidney Translational Research Center (KTRC) for kidney samples and the HuBMAP grant U54HL145608. We thank Dr. E. Bigio and Dr. M.-M. Mesulam from Mesulam Center for Cognitive Neurology and Alzheimer's Disease (MCCNAD) for providing the de-identified autopsy brain samples; and MCCNAD is supported by NIH P30 AG013854. Work done in D.S.K. laboratory is supported by NEI P30 grant, P30EY034070-01 and in part by an unrestricted grant from Research to Prevent Blindness awarded to the Gavin Herbert Eye Institute.

## Author contributions

L.S. conceived the idea and designed the project; W.Z. developed PRM-SRS algorithm and coded it with help from L.S., Z.L. and A.A.F.; Y.L. and A.A.F. carried out the imaging experiments and collected data with the help from L.S., Z.L. and H.Z.; A.A.F., L.S., H.J., Y.L. and Z.L. analyzed the image and generated the figures; J.Y.W. helped with project design; X.C. prepared human cell and brain tissue samples; S.J. contributed to data interpretations for human kidney samples. J.Y. and H.S. contributed to data interpretations for mouse brain samples. F.G. and D.S.K. conducted

mass spectrometry measurement; A.A.F. and L.S. drafted and revised the manuscript with the input from all other authors.

## Competing interests

The authors declare no competing interests.
