## [Peer Review File · Nature Communications]

Multi-molecular hyperspectral PRM-SRS microscopyREVIEWER COMMENTS

Reviewer #1 (Remarks to the Author):

In this manuscript, Wenxu Zhang and colleagues report on a non-destructive and label-free optical imaging method, called Penalized Reference Matching algorithm with Stimulated Raman Scattering (PRM-SRS) microscopy, for distinguishing various lipid subtypes in cells and tissues. This approach appears to allow for direct visualization and identification of a variety of lipid species in situ with relatively high chemical specificity and cellular/subcellular (organelle) resolution. I cannot comment on the new hyperspectral imaging platform described in the manuscript in regards to development, application, accuracy, or chemical analysis. However, the output (in situ visualization/identification of lipid species across taxa) is impressive and analogous to other more destructive methods. The authors show that the PRM-SRS approach can identify a variety of lipid subtypes, from bulk energy substrates (such as triglycerides) to esterified cholesterol to mitochondrial phospholipids (cardiolipin) with excellent subcellular resolution in a variety of cells and tissues with different challenges to imaging (for example, sectioned mouse tissue versus whole mount insect tissue). The PRM-SRS lipid analysis appears analogous to other destructive methods, such as labeled dye imaging, NMR, and Gas Chromatography Mass Spectrometry. This includes previously published work as well as analyses performed in this manuscript (for example, highlighting that both GCMS and PRM-SRS imaging show changes in cholesterol /phosphatidylethanolamine (PE) ratios in old mouse hippocampal sections). Furthermore, analysis of the insect (*Drosophila melanogaster*) fat body with PRM-SRS highlights typical large lipid droplets (stored triglycerides) and distinct lysosomes, as previously described, with excellent subcellular resolution. The mapping of cholesterol in young and old fat body samples also correlates well with previously published data.

In summary, this new approach appears to provide an excellent platform for label-free optical imaging of a variety of lipid species, in situ, that will enhance current lipidomic and label-dependent assays used to explore physiology and pathophysiology associated with changes in lipid biology.

Reviewer #2 (Remarks to the Author):

In this manuscript the authors discuss an algorithm that helps identify component spectra from a Raman spectrum consisting of multiple components. The algorithm makes use of reference spectra and the spectral filtering is enabled by a scoring procedure that makes use of a penalty coefficient that tracks spectral alignment of the component spectra with a given reference. This analysis approach is tested on Raman spectra of lipid standards and on SRS images of a variety of tissue sample. The overall conclusion is that the described analysis approach performs well and is able to quantitatively determine the relative amount of the target compound that contributes to the ensemble spectra.

The manuscript is well written and the procedures are clearly explained. The proof-of-principle studies demonstrate that the method has merit. In general, 'fishing' out target compounds from ensemble Raman spectra is a challenge, and the current method appears to be performing quite well. In this sense, the work can be of use to the Raman community, especially when it concerns lipid analysis.

At the same time, the novelty of the work is the algorithm itself. While certainly interesting and useful, referencing approaches for spectral analysis are not new and the current method can be considered a variation on a known concept. The application of the penalty coefficient is a good strategy, but the overall approach is rather empirical in nature (adjusting penalty scores till it behaves well) and not grounded in a fundamentally new principle. Therefore, the current method is considered an interesting variation on a known strategy rather than breakthrough advance. For this reason, the present manuscript is not an immediate match for Nature Communications.

That said, the method performs well and is, therefore, of value to those interested in Raman-based lipid analysis of biological samples. In order to improve the manuscript, the following points are noted:

- 1) The authors keep repeating that their method is quantitative. However, in Figure 1D of the supplementary material it is clear that the method is unable to extract quantitatively correct values from pixels that contain lower quantities of the target compound against a

larger background. In general, the paper misses a discussion on actual sensitivity. It is easy to detect target compounds from a binary (or ternary) mixtures, but in biological samples the pixel spectra are typically composed of hundreds (or many more!) compounds. Using more expansive controls in different concentration regimes, the paper should clearly state for what concentrations the extracted numbers make sense, and when they start to become inaccurate.

2) In Figure 7K, the authors perform a nice comparison between PRM-SRS and GC/MS. The data shows good agreement. Nonetheless, this is only one comparison. Similar to point 1), it would be good to show more examples to demonstrate good (quantitative) performance over a wider range of samples.

3) In Figure 4, there is reasonable overlap between the fluorescence and the PRM-SRS image of CL in cells. Figures 4C and 4D show a similarity analysis averaged over the entire image. Nonetheless, there are also differences as the spatial overlap is clearly not perfect (the fluorescence image in 4B is very dim, which makes comparing these images quite difficult). The differences should be discussed.

4) Analysis is only performed in the CH-stretching region of the spectrum. A stronger case for the method would be made if imaging results were also performed in the fingerprint region.

5) Examples are exclusively focused on lipids, thus limiting the generality of the analysis approach. Could the authors include an example of another molecular class (carbohydrate, nucleic acids etc.)?

Reviewer #3 (Remarks to the Author):

The manuscript developed a new Penalized Reference Matching (PRM) algorithm to detect specific lipids in a hyperspectral Raman image. The algorithm was tested on Stimulated Raman Scattering (SRS) microscopy images of kidney, mice hippocampus, and human brain samples. Using the PRM-SRS method, the authors demonstrated the spatial mapping of various lipids such as cholesterol, cholesterol ester, phosphatidylethanolamine (PE), sphingosine, and cardiolipin. Although interesting, the manuscript could be improved by incorporating the following comments.

1. The specificity of identifying various lipid species in the high wavenumber Raman spectral

region (2700 – 3100 cm^{-1}) is not convincing. For example, cholesterol ester could be distinguished from cholesterol with high specificity using the ester group Raman band (C=O bond vibration) at around 1739 cm^{-1} . Why not use the Raman fingerprint region for some SRS applications? The authors should prepare a table for the 38 biomolecules used in the manuscript and compare the peak position in the spectral region of 2700 – 3100 cm^{-1} to show that the peak positions are indeed different for each of the molecules.

2. How is spectral preprocessing performed in Figure 3A? Why do some of the Raman spectra of old mice samples have negative values?

3. Figures 3C and 3D seem to be normalized to 1, whereas Figures 3A and 3B are not normalized to 1. Will this affect the similarity score?

4. How the algorithm will distinguish cholesterol from other fatty acid signatures is unclear. For example, the peaks shown in Figures 3A and 3B are similar in the Raman spectrum of other fatty acids and may not be specific to cholesterol, as claimed by the authors. The same problem exists for TAG, CL, and PE. The Raman spectrum of fatty acids will have similar features to that of TAG, CL, and PE. It's essential to understand how the algorithm distinguishes the spectral characteristics of specific lipids.

5. Potential interference from proteins and nucleic acids further complicates the Raman data analysis. The manuscript is not exploring the interference from the Raman features of proteins, nucleic acids, and carbohydrates on the lipid similarity scores.

6. In Figure 4, two-photon fluorescence microscopy imaging was used to validate the SRS imaging of cardiolipin (CL). However, the authors should consider a more quantitative method, such as GC-MS or LC-MS, to validate the results.

7. In Figure 7L, some of the similarity scores are greater than 1. What is the significance of this, and why is it greater than 1?

8. Although the underlying biology may not be the focus of this study, the authors should consider validating the lipid up/down-regulation with the corresponding transcriptomic profile and the associated proteins to provide more confidence to the results.

9. Why was this particular ceramide subspecies (C12 Cer (d18:1/12:0)) measured? What is the significance of this ceramide?

10. For supplementary figure S1, which subspecies of TAG and Ceramide were considered?

11. In Figure 7K, what is the definition of "Ratio of Simplex-Normalized % Composition"?

Some example calculations should be shown. Since, Cholesterol, PC, and PE will have

different subspecies, how was it calculated in GC-MS and SRS?

12. It's not clear how Figure 8A was created. Which Raman spectral band of Sphingosine was utilized?

13. The authors should clearly describe that Figure 8E is for the ratio in Figure 8D. It's not clear from the description. Also, which dotted line is referred to in Figure 8F?

REVIEWER COMMENTS

Reviewer #1 (Remarks to the Author):

In this manuscript, Wenxu Zhang and colleagues report on a non-destructive and label-free optical imaging method, called Penalized Reference Matching algorithm with Stimulated Raman Scattering (PRM-SRS) microscopy, for distinguishing various lipid subtypes in cells and tissues. This approach appears to allow for direct visualization and identification of a variety of lipid species in situ with relatively high chemical specificity and cellular/subcellular (organelle) resolution. I cannot comment on the new hyperspectral imaging platform described in the manuscript in regards to development, application, accuracy, or chemical analysis. However, the output (in situ visualization/identification of lipid species across taxa) is impressive and analogous to other more destructive methods. The authors show that the PRM-SRS approach can identify a variety of lipid subtypes, from bulk energy substrates (such as triglycerides) to esterified cholesterol to mitochondrial phospholipids (cardiolipin) with excellent subcellular resolution in a variety of cells and tissues with different challenges to imaging (for example, sectioned mouse tissue versus whole mount insect tissue). The PRM-SRS lipid analysis appears analogous to other destructive methods, such as labeled dye imaging, NMR, and Gas Chromatography Mass Spectrometry. This includes previously published work as well as analyses performed in this manuscript (for example, highlighting that both GCMS and PRM-SRS imaging show changes in cholesterol /phosphatidylethanolamine (PE) ratios in old mouse hippocampal sections). Furthermore, analysis of the insect (*Drosophila melanogaster*) fat body with PRM-SRS highlights typical large lipid droplets (stored triglycerides) and distinct lysosomes, as previously described, with excellent subcellular resolution. The mapping of cholesterol in young and old fat body samples also correlates well with previously published data.

In summary, this new approach appears to provide an excellent platform for label-free optical imaging of a variety of lipid species, in situ, that will enhance current lipidomic and label-dependent assays used to explore physiology and pathophysiology associated with changes in lipid biology.

We appreciate your positive comments.

Reviewer #2 (Remarks to the Author):

In this manuscript the authors discuss an algorithm that helps identify component spectra from a Raman spectrum consisting of multiple components. The algorithm makes use of reference spectra and the spectral filtering is enabled by a scoring procedure that makes use of a penalty coefficient that tracks spectral alignment of the component spectra with a given reference. This analysis approach is tested on Raman spectra of lipid standards and on SRS images of a variety of tissue sample. The overall conclusion is that the described analysis approach performs well and is able to quantitatively determine the relative amount of the target compound that contributes to

the ensemble spectra.

The manuscript is well written and the procedures are clearly explained. The proof-of-principle studies demonstrate that the method has merit. In general, 'fishing' out target compounds from ensemble Raman spectra is a challenge, and the current method appears to be performing quite well. In this sense, the work can be of use to the Raman community, especially when it concerns lipid analysis.

At the same time, the novelty of the work is the algorithm itself. While certainly interesting and useful, referencing approaches for spectral analysis are not new and the current method can be considered a variation on a known concept. The application of the penalty coefficient is a good strategy, but the overall approach is rather empirical in nature (adjusting penalty scores till it behaves well) and not grounded in a fundamentally new principle. Therefore, the current method is considered an interesting variation on a known strategy rather than breakthrough advance. For this reason, the present manuscript is not an immediate match for Nature Communications.

That said, the method performs well and is, therefore, of value to those interested in Raman-based lipid analysis of biological samples. In order to improve the manuscript, the following points are noted:

1) The authors keep repeating that their method is quantitative. However, in Figure 1D of the supplementary material it is clear that the method is unable to extract quantitatively correct values from pixels that contain lower quantities of the target compound against a larger background. In general, the paper misses a discussion on actual sensitivity. It is easy to detect target compounds from a binary (or ternary) mixtures, but in biological samples the pixel spectra are typically composed of hundreds (or many more!) compounds. Using more expansive controls in different concentration regimes, the paper should clearly state for what concentrations the extracted numbers make sense, and when they start to become inaccurate.

Response: Thank you for the helpful comments. We agree that this method has limitations especially in the case of low concentration components. As presented in Figure S1D, when the target molecule is dominant in the region of interest, the PRM-SRS shows robust and quantitative results. Many factors may affect the results. For example, Raman spectra of the same lipid subtype can be different in different physical and chemical conditions, such as in rigid gel phase and liquid phase, because of the flexibility of carbon chains. We revised the manuscript in Line 104 as below.

In this study, we focus on using dominant components to illustrate the application of PRM-SRS in analyzing different lipid subtypes. In future follow-up study, we plan to further enhance the detection sensitivity to improve the signal-to-noise ratio for examining molecules of low abundance.

2) In Figure 7K, the authors perform a nice comparison between PRM-SRS and

GC/MS. The data shows good agreement. Nonetheless, this is only one comparison. Similar to point 1), it would be good to show more examples to demonstrate good (quantitative) performance over a wider range of samples.

Response: We appreciate your comments. Indeed, we have used a range of different samples, from cultured cells to tissue samples and from *Drosophila*, mouse brain to human samples. In murine brain samples, we have compared GC-MS with PRM-SRS. We also used knock-down of an enzyme critical for cardiolipin biosynthesis to show the reduction in specific lipid (cardiolipin) signal in PRM-SRS, consistent with that measured by NAO fluorescence imaging.

3) In Figure 4, there is reasonable overlap between the fluorescence and the PRM-SRS image of CL in cells. Figures 4C and 4D show a similarity analysis averaged over the entire image. Nonetheless, there are also differences as the spatial overlap is clearly not perfect (the fluorescence image in 4B is very dim, which makes comparing these images quite difficult). The differences should be discussed.

Response: Thank you for the helpful comments, we have added the following discussion to the manuscript Line 243 as below.

To compare the image similarity between fluorescence image and PRM-SRS image, similarity index and normalized mean squared error were calculated (Fig. S6). We found that the similarity index between fluorescence images of NAO stained cardiolipin and PRM-SRS images of cardiolipin was higher than other lipid subtypes. Normalized mean squared error for cardiolipin is lower than that of other lipid subtypes, supporting the highest similarity between NAO stained cardiolipin and PRM-SRS image of cardiolipin. Importantly, down-regulating PGS1, an enzyme critical for cardiolipin biosynthesis, significantly reduced NAO-staining signal and PRM-SRS measured cardiolipin signal, supporting that PRM-SRS measured cardiolipin signals reflect the cardiolipin levels in the samples. These results of similarity comparison show that PRM-SRS describes well the cardiolipin distribution.

Fig. S6. Similarity indices and normalized mean squared errors between NAO stained cardiolipin image and PRM-SRS detected lipid subtype images (cardiolipin, PE, cholesterol, and sphingosine). In the two samples (a: Control, b: ShPGS1), the similarity index of cardiolipin was higher than the indices of other lipid subtypes. Normalized mean squared error of cardiolipin image was lower than other lipid subtypes. Due to the low intensity in ShPGS1 sample, similarity index was lower than control, and normalized mean squared error was higher than control.

4) Analysis is only performed in the CH-stretching region of the spectrum. A stronger case for the method would be made if imaging results were also performed in the fingerprint region.

Response: Thank you for the important suggestion. The spectral information in fingerprint region is very important to understand specific functional group distributions and protein secondary structures. However, in lipid subtype analysis, carbon chain vibration signal is critical information, and the vibration of the chains can be affected by head group difference (1). In addition, in Raman spectra, due to the strong polarizability changes, the CH-stretching region has much stronger signal than the fingerprint region. Therefore, we focused on the CH-stretching region only to analyze lipid subtype distribution. In our future study applying PRM-SRS to visualize protein distribution, the fingerprint region will be examined.

To clarify this point, we revised the manuscript (Line 498) as follows.

In this study, due to the stronger signal of the CH-stretching region than the fingerprint region, we focused on the CH-stretching region to analyze lipids. In our future study applying PRM-SRS to visualize protein distribution, the fingerprint region will be a focus of our attention.

5) Examples are exclusively focused on lipids, thus limiting the generality of the analysis approach. Could the authors include an example of another molecular class (carbohydrate, nucleic acids etc.)?

Response: Thank you for the constructive discussion. Actually, the other molecular

groups of carbohydrates, nucleic acids, proteins, and other small molecules can be studied to reveal specific biological mechanisms use the same approach. But, in this study, we narrowed down the scope of PRM-SRS analysis and focused only on demonstrating results of lipid subtypes. In our future study, we will expand the spectral range to other more biomolecular subtypes.

Reviewer #3 (Remarks to the Author):

The manuscript developed a new Penalized Reference Matching (PRM) algorithm to detect specific lipids in a hyperspectral Raman image. The algorithm was tested on Stimulated Raman Scattering (SRS) microscopy images of kidney, mice hippocampus, and human brain samples. Using the PRM-SRS method, the authors demonstrated the spatial mapping of various lipids such as cholesterol, cholesterol ester, phosphatidylethanolamine (PE), sphingosine, and cardiolipin. Although interesting, the manuscript could be improved by incorporating the following comments.

1. The specificity of identifying various lipid species in the high wavenumber Raman spectral region (2700 – 3100 cm^{-1}) is not convincing. For example, cholesterol ester could be distinguished from cholesterol with high specificity using the ester group Raman band (C=O bond vibration) at around 1739 cm^{-1} . Why not use the Raman fingerprint region for some SRS applications? The authors should prepare a table for the 38 biomolecules used in the manuscript and compare the peak position in the spectral region of 2700 – 3100 cm^{-1} to show that the peak positions are indeed different for each of the molecules.

Response: Thank you for the critical comment. Raman scattering intensity is proportional to polarizability change of the vibrational mode. Molecular volume is one of the critical parameters related to polarizability change, so Raman scattering intensity in -CH vibration region is stronger than fingerprint region (1). In addition, because of carbon chain of lipids, -CH vibration signal is important to understand lipid structures. Therefore, in this paper, we focused on -CH vibration signal with lipid subtype distributions. Table S2 about Gaussian fitting parameters to characterize spectra of 38 lipid subtypes is added to the manuscript, and using t-SNE plot (Figure S1), capability to differentiate the lipid subtypes based on CH stretching spectra. In addition, from the inner product table (Table S1) of 8 different lipid subtypes, the potential of PRM-SRS to quantitatively analyze dominant molecules is presented.

Figure S1: CH stretching region spectrum and t-SNE plot of their fitting parameters. A) Spontaneous Raman spectrum of PE lipid. With four Gaussian peaks centered on four different wavenumbers, spectral shape was fitted. The fitting parameters, sigma and amplitude of each Gaussian peak were summarized in Table S2. **B)** To show the capability of lipid subtype separation based on CH stretching region spectra, t-SNE plot of 38 lipid subtypes is presented. The plot shows enough distances between each lipid subtype.

	C12 Ceramide	PE	PC	Cholesterol	Cholesterol Ester	TAG	Cardiolipin	Sphingosine
C12 Ceramide	1	0.9396	0.9504	0.9324	0.9415	0.8536	0.9229	0.9927
PE	0.9396	1	0.9953	0.973	0.9854	0.9497	0.9932	0.9155
PC	0.9504	0.9953	1	0.9728	0.9851	0.9437	0.9904	0.9277
Cholesterol	0.9324	0.973	0.9728	1	0.9941	0.9658	0.9725	0.8978
Cholesterol Ester	0.9415	0.9854	0.9851	0.9941	1	0.9604	0.9838	0.9112
TAG	0.8536	0.9497	0.9437	0.9658	0.9604	1	0.9638	0.8083
Cardiolipin	0.9229	0.9932	0.9904	0.9725	0.9838	0.9638	1	0.8903
Sphingosine	0.9927	0.9155	0.9277	0.8978	0.9112	0.8083	0.8903	1

Table S1: Inner product values of 8 different lipid subtypes. The inner products of 8 different lipid subtypes show the capability of quantitative analysis. The lipid subtype pairs having high structural similarities, for example, PC and PE show very high inner product value. On the contrary, the lipid subtype pairs having low structural similarities, such as, Cholesterol and Sphingosine, have lower inner product values than other lipid subtype pairs.

	PE	PLS	Cholesterol	Cardiolipin	Sphingosine	Cholesterol Ester	DsgPI	LaPI	LaPG	LPA
2840_Amp	0.646422	0.210139	0.093545	0.385544	0.407072	0.534452	0.637567	0.518231	0.484989	0.478832
2840_Sig	3.571	2.698	11.88	3.622	5.623	8.614	3.517	3.865	4.113	3.869
2880_Amp	1	1	1	1	1	1	1	1	1	1
2880_Sig	20.31	20.86	19.05	21	17.91	19.31	20.76	20.53	20.15	20.09
2940_Amp	0.710729	1.724821	1.344092	0.785591	0.263582	0.925578	0.643712	0.572187	0.535593	0.4749
2940_Sig	19.21	18.94	20.16	18.02	10.85	19.1	21.09	17.48	17.59	19.73
3005_Amp	0.170593	0.196365	0	0.312737	0.018962	0.055071	0.121273	0.158748	0.133565	0.198126
3005_Sig	11.62	11.96	0	10.38	30	17.55	8.622	10.89	11.78	6.676

	PC	TAG	Lip_16:0	Lip_18:0	DHA_Omega_3-22-6	Omega_3-25-5	Cera_C24	Cera_C22	Cera_C24-1	PC_18:1
2840_Amp	0.616895	0.246094	0.735662	0.753657	0.004776	0.159545	0.597738	0.819062	0.610301	0.501554
2840_Sig	3.654	3.817	5.974	5.948	30	2.91	4.924	6.785	5.097	3.964
2880_Amp	1	1	1	1	1	1	1	1	1	1
2880_Sig	21.01	17.49	6.822	7.126	23.97	22.98	17.43	7.204	18.64	22.66
2940_Amp	0.662991	1.386402	0.242621	0.250994	1.005015	0.759629	0.262598	0.237303	0.386831	0.614221
2940_Sig	19.06	16.33	21.9	22.24	14.33	14.32	10.52	20	12.38	21.64
3005_Amp	0.218146	0.145165	0	0	1.124904	0.67571	0.028397	0.00704	0.130717	0.170051
3005_Sig	17.26	14.69	0	0	17.23	15.21	9.294	3.787	16.41	12.79

	PE_18:1	Cera_C18-1	Deox_Cera_C24-1	Deox_Cera_C16	Deox_DiHy_Cera_C14	Deox_DiHy_Cera_C24-1	Lip_24:5	LysoPA	LysylDG	PS
2840_Amp	0.488277	0.287382	0.789593	0.500119	0.500119	0.494755	0.159589	0.594091	0.622972	0.582002
2840_Sig	3.79	13.12	5.972	4.078	4.078	3.989	2.909	4.935	4.994	3.534
2880_Amp	1	1	1	1	1	1	1	1	1	1
2880_Sig	22.31	18.67	8.196	16.68	16.68	16.84	22.99	15	15.58	19.61
2940_Amp	0.571619	0.408259	0.252	0.266428	0.266428	0.257381	0.759561	0.284488	0.493822	0.569123
2940_Sig	21.81	11.81	22.69	13.8	13.8	12.12	14.32	15.52	18.59	17.91
3005_Amp	0.197337	0.110807	0.038711	0.07336	0.07336	0.055314	0.675759	0.019894	0.04365	0.211444
3005_Sig	5.941	11.75	3.948	4.338	4.338	2.738	15.21	5.417	14.37	12.85

	TAG_16:0	TAG_18:1	DAG_16:0	DAG_18:0_24:0	HeLa_Lysosome	Fatbody_Lysosome	Cera_C12	CDPDG
2840_Amp	0.737984	0.586962	0.75	0.506423	0.347758	0.42752	0.287382	0.709225
2840_Sig	5.974	3.653	5.447	3.347	5	3.304	13.12	4.886
2880_Amp	1	1	1	1	1	1	1	1
2880_Sig	8.857	20.78	9.169	20.96	20.96	20.4	18.67	14.39
2940_Amp	0.277126	0.633666	0.234547	0.697421	0.868214	0.895266	0.408259	0.484391
2940_Sig	20.63	16.35	22.04	16.03	18.49	17.08	11.81	20.63
3005_Amp	0	0.16755	0	0.241491	0.177682	0.14729	0.110807	0.008947
3005_Sig	0	5.325	0	11.2	9.951	9.398	11.75	3.867

Table S2: Fitting parameters of four Gaussian curves to describe spectral shape. Fitting parameters to define shapes of four Gaussian peaks, amplitudes and sigma values, are listed. Based on this information t-SNE plot was prepared.

2. How is spectral preprocessing performed in Figure 3A? Why do some of the Raman spectra of old mice samples have negative values?

Response: Thank you for bringing this to our attention. SRS signals cannot be negative, but because of noise and the background subtraction process, the spectra can have negative values, which are individually corrected later as a preprocessing step. We have clarified this in the methods section, and have corrected the figure to show final preprocessed spectra just before the similarity score calculation steps.

3. Figures 3C and 3D seem to be normalized to 1, whereas Figures 3A and 3B are not normalized to 1. Will this affect the similarity score?

Response: Figure 3 describes the process of acquiring a Raman spectrum and comparing it to a reference standard spectrum. Figures 3A and 3B are raw data, and Figures 3C and 3D are normalized data of a reference spectrum. In the code of PRM-SRS, every spectrum needs to be normalized before computing similarity. Without this

critical normalization step, it would be as if we compared the fatbody spectra to a horizontal line, resulting in a nearly 0 similarity score.

4. How the algorithm will distinguish cholesterol from other fatty acid signatures is unclear. For example, the peaks shown in Figures 3A and 3B are similar in the Raman spectrum of other fatty acids and may not be specific to cholesterol, as claimed by the authors. The same problem exists for TAG, CL, and PE. The Raman spectrum of fatty acids will have similar features to that of TAG, CL, and PE. It's essential to understand how the algorithm distinguishes the spectral characteristics of specific lipids.

Response: Thank you for bringing this to our attention. The multivariate aspect of this analysis is key. Even though the structures of fatty acids, and consequently their Raman spectra, are similar, there are differences that we can exploit to measure and visualize relative concentration and spatial differences, even if they're small. With respect to the lipids listed above, a discriminating feature between PE, TAG, and CL is the head group. Depending on the headgroup structure, Raman spectra of each lipid subtype can be varied. Additionally, considering the number of -CH bond signal in -CH vibration region is more related to the fatty acid concentration, but structural variance in head groups induce energy difference in vibrational mode of fatty acids, see reference (1). Cholesterol also has -CH bonds. Therefore, as shown in Table 2, the molecule has its own -CH stretching Raman signal. Due to cholesterol's unique structure, it is easier to distinguish the molecule than other phospholipids having similar structures, for example, PC and PE (See Table 1). In summary, even if reference spectra are similar, such as TAG, Cholesterol, and Esterified Cholesterol, if there are any differences at all, there will be differences in the similarity score result. Other methods such as pseudo-inverse matrix multiplication have been published in other hyperspectral applications, but have their own limitations. We have added a new supplementary figure S4 that compares these in greater detail. With proper bit-depth, these small differences in Raman spectra and their resulting similarity scores can be captured.

5. Potential interference from proteins and nucleic acids further complicates the Raman data analysis. The manuscript is not exploring the interference from the Raman features of proteins, nucleic acids, and carbohydrates on the lipid similarity scores.

Response: Thank you for the constructive discussion. Indeed, the potential interference from other biomolecules, including proteins, nucleic acids and carbohydrates is a very important issue when imaging lipids. Our method focuses on the dominant component of each pixel and signals from non-dominant molecular species are filtered out, thereby minimizing the influence by such non-dominant species. We have addressed this in the manuscript Line 104 as below.

In this study, we focus on using dominant components to illustrate the application of PRM-SRS in analyzing different lipid subtypes. In future follow-up study, we plan to further enhance the detection sensitivity to improve the signal-to-noise ratio for examining molecules of low abundance.

6. In Figure 4, two-photon fluorescence microscopy imaging was used to validate the

SRS imaging of cardiolipin (CL). However, the authors should consider a more quantitative method, such as GC-MS or LC-MS, to validate the results.

Response: Thank you for the suggestion. We agree that GC-MS or LC-MS may provide additional validation and we have incorporated GC-MS validation in Fig 7. In this study, we focus on developing the SRS-based imaging method and analysis of cardiolipin. We have used knocking-down of PGS1, an enzyme critical for cardiolipin synthesis, to demonstrate the specific signal changes in PRM-SRS associated with cardiolipin change.

7. In Figure 7L, some of the similarity scores are greater than 1. What is the significance of this, and why is it greater than 1?

Response: Figure 7L describes the ratio of similarity scores between two lipid subtypes, so the number that is greater than 1 is related to the information about which lipid subtype is dominant.

8. Although the underlying biology may not be the focus of this study, the authors should consider validating the lipid up/down-regulation with the corresponding transcriptomic profile and the associated proteins to provide more confidence to the results.

Response: Thank you for the suggestion. We plan to incorporate this in our follow-up study. We added this into the Discussion (Line 510).

This new platform can also be applied to studying metabolism of diverse types of biomolecules in cell and tissue samples. For example, when combined with transcriptomics analyses following up- or down-regulation of lipid metabolism genes, it will be highly useful for investigating metabolic changes under different pathophysiological conditions.

9. Why was this particular ceramide subspecies (C12 Cer (d18:1/12:0)) measured? What is the significance of this ceramide?

Response: Ceramides play crucial roles in various biological signaling pathways, and their functions can vary depending on the length of their fatty acid chains (2-4). While extensive research has explored the roles of ceramides with long chain fatty acids(2), those with shorter chain lengths have received less attention. Lipid subtypes characterized by shorter chain lengths exhibit relatively higher fluidity. Consequently, in our quest to comprehend how these molecules behave, it is imperative to recognize the significance of ceramides with short chain lengths, like C12, alongside their counterparts with different chemical and physical characteristics.

10. For supplementary figure S1, which subspecies of TAG and Ceramide were considered?

Response: As described in the figure legend, the samples are mixtures of ceramides and TAG. We did not analyze different subspecies of TAG of Ceramide, since our

method does not have enough resolution to distinguish different subspecies of these lipids.

11. In Figure 7K, what is the definition of “Ratio of Simplex-Normalized % Composition”? Some example calculations should be shown. Since, Cholesterol, PC, and PE will have different subspecies, how was it calculated in GC-MS and SRS?

Response: All lipid subtype concentrations in pmol from GC-MS sum to 100%. The amount of a lipid subtype is calculated by summing the pmol values of the individual lipid subtypes. The amount of the lipid subtype is then normalized to total lipid content. The process to calculate the contents is as described here, but denominators of the two normalized compositions are the same, total lipid amount. Accordingly, the “Ratio of Simplex-Normalized % Composition” should correspond to the ratio of two lipid subtypes. We have revised the subtitle in the figure to make it clearer.

12. It’s not clear how Figure 8A was created. Which Raman spectral band of Sphingosine was utilized?

Response: Figure 8A and 8B are similarity score images using PRM-SRS. Each pixel intensity is the similarity score result from PRM (with a range from 0 to 1), not the intensity of a single Raman peak. Similarity scores between measured spectra and reference spectra of Sphingosine and Cardiolipin were measured using PRM-SRS pixel by pixel. The analyzed spectral region is the CH vibration region. The entire imaging and analysis process is exactly the same as that described for Fig. 2 and Fig. 3, except that reference spectra of sphingosine and cardiolipin were used.

13. The authors should clearly describe that Figure 8E is for the ratio in Figure 8D. It’s not clear from the description. Also, which dotted line is referred to in Figure 8F?

Response: We have revised Fig. 8E caption to make it clearer.

References

1. Czamara K, Majzner K, Pacia MZ, Kochan K, Kaczor A, Baranska M. Raman spectroscopy of lipids: a review. *Journal of Raman Spectroscopy*. 2015;46(1):4-20.
2. Liang L, Li D, Zeng R, Zhang H, Lv L, Wei W, et al. Long- and very long-chain ceramides are predictors of acute kidney injury in patients with acute coronary syndrome: the PEACP study. *Cardiovascular Diabetology*. 2023;22(1):92.
3. Li PL, Zhang Y. Cross talk between ceramide and redox signaling: implications for endothelial dysfunction and renal disease. *Handb Exp Pharmacol*. 2013(216):171-97.
4. Basnakian AG, Ueda N, Hong X, Galitovsky VE, Yin X, Shah SV. Ceramide synthase is essential for endonuclease-mediated death of renal tubular epithelial cells induced by hypoxia-reoxygenation. *Am J Physiol Renal Physiol*. 2005;288(2):F308-14.

REVIEWERS' COMMENTS

Reviewer #2 (Remarks to the Author):

I have read the received manuscript and the response by the authors to the Reviewers' comments. My previous concerns have been largely addressed. I have no further concerns.

Reviewer #3 (Remarks to the Author):

The authors have revised the manuscript according to the comments.